# Re-expression of CA1 and entorhinal activity patterns preserves temporal context memory at long timescales

Futing Zou [1] ✉, Guo Wanjia [1], Emily J. Allen [2], Yihan Wu[3], Ian Charest [4], Thomas Naselaris[5], Kendrick Kay[2], Brice A. Kuhl [1,6], J. Benjamin Hutchinson [1,7] & Sarah DuBrow [1,6,7,8]

Converging, cross-species evidence indicates that memory for time is supported by hippocampal area CA1 and entorhinal cortex. However, limited evidence characterizes how these regions preserve temporal memories over long timescales (e.g., months). At long timescales, memoranda may be encountered in multiple temporal contexts, potentially creating interference. Here, using 7T fMRI, we measured CA1 and entorhinal activity patterns as human participants viewed thousands of natural scene images distributed, and repeated, across many months. We show that memory for an image's original temporal context was predicted by the degree to which CA1/entorhinal activity patterns from the first encounter with an image were re-expressed during re-encounters occurring minutes to months later. Critically, temporal memory signals were dissociable from predictors of recognition confidence, which were carried by distinct medial temporal lobe expressions. These findings suggest that CA1 and entorhinal cortex preserve temporal memories across long timescales by coding for and reinstating temporal context information.

Episodic memory fundamentally involves the ability to remember not only *what* happened in the past, but *when* it happened[1]. Indeed, placing memories in time critically enables experiences to be organized into personal narratives that span weeks, months, and years[2]. Yet, the majority of cognitive neuroscience studies of human memory only consider memory across relatively short timescales (overwhelmingly, within a single experimental session/day). At longer timescales, one of the particular challenges to retaining precise temporal memories is that previously-encoded information is likely to be 'reencountered' in new temporal contexts[3]. For example, remembering precisely when you *first saw* a particular movie may be complicated by re-watching that movie at a later date. Understanding how memories of specific temporal contexts are preserved when experiences are repeated over

long timescales (days, weeks, months) requires identifying not only the neural structures that are involved, but the mechanistic contributions that these structures support.

Broadly, the medial temporal lobe (MTL) system is known to critically support episodic memory[4–6]. However, within the MTL system, hippocampal subfield CA1 and entorhinal cortex (ERC) have emerged as being particularly important for processing and remembering temporal information[7–11]. For example, so-called "time cells" in CA1 and "ramping cells" in ERC have been shown to code for elapsed time in rodents[12–15], with similar effects recently observed in the human hippocampus and ERC[16,17]. Importantly, although individual time cells typically operate over very short timescales (seconds), ensembles of time cells may provide *temporal context representations* that span

[1]Department of Psychology, University of Oregon, Eugene, OR, USA. [2]Center for Magnetic Resonance Research, Department of Radiology, University of Minnesota, Minneapolis, MN, USA. [3]Graduate Program in Cognitive Science, University of Minnesota, Minneapolis, MN, USA. [4]Department of Psychology, University of Montreal, Montreal, QC, Canada. [5]Department of Neuroscience, University of Minnesota, Minneapolis, MN, USA. [6]Institute of Neuroscience, University of Oregon, Eugene, OR, USA. [7]These authors jointly supervised this work: J. Benjamin Hutchinson, Sarah DuBrow. [8]Deceased: Sarah DuBrow. ✉e-mail: futingz@uoregon.edu

much longer timescales[18] and allow individual memories to be 'placed' in time[19]. These temporal context representations are thought to integrate information about temporally adjacent events as well as ongoing internal operations or processes[20]. While human fMRI studies have provided important evidence that activation levels in the hippocampus and ERC are associated with the precision of temporal memory[21,22], measures of activation, alone, are not able to capture the ensemble representations in which temporal context information is thought to be encoded[23,24].

Importantly, to the extent that CA1 and ERC do code for the temporal context in which events occur, there are multiple—and mechanistically distinct—ways in which these representations might preserve temporal memories. On the one hand, when a given stimulus is re-encountered in a new temporal context, CA1 and/or ERC may encode the new temporal context as *distinct* from the original context[25]. Forming distinct temporal context representations across repeated encounters is potentially beneficial to temporal memory by improving discriminability of these contexts[26]. On the other hand, when a stimulus is re-encountered in a new temporal context, this potentially creates an opportunity to *reinstate* the prior temporal context[20,27]. For example, when a familiar movie is on television, this might trigger recall of the original temporal context in which the movie was encountered. Critically, this reinstatement should strengthen the association between the movie and its original temporal context[28]. According to leading theoretical accounts, a stronger association between a given memory (e.g., the movie) and a particular temporal context (e.g., the movie's original temporal context) will directly support the ability to place that memory in time[29]. Thus, in contrast to a context distinctiveness account, a context reinstatement account makes the prediction that, when stimuli are re-encountered, memory for the original temporal context will be preserved to the extent that activity patterns in CA1 and/or ERC are *similar* to (or reinstate) the activity patterns expressed when the stimulus was first encountered.

Here we sought to characterize the neural mechanisms that preserve temporal context memory when events are re-encountered across long timescales (days to months). To address this, we describe a massive human fMRI experiment in which participants encountered thousands of natural scene images repeatedly during 30–40 scan sessions distributed over an 8–10 month window[30]. After all scans were completed, participants performed a temporal memory task in which a subset of images were presented and participants were asked to estimate when each image was *first encountered* (on a scale that ranged from days to months in the past). The focus of our analyses was to test whether temporal memory precision was predicted by the degree to which patterns of neural activity expressed when images were first encountered were re-expressed when these images were re-encountered (a potential marker of context reinstatement). By leveraging the ultra-high field strength (7 T) and high spatial resolution (1.8-mm) of our imaging protocol, we interrogated subregions of the hippocampus (including CA1) and surrounding MTL structures (including ERC). This experimental design yielded an unprecedented ability to understand how temporally-precise memories are preserved over long timescales that are critical for real-world memories.

## Results

### Precise temporal memory persists across months

Eight participants completed two experimental phases (Fig. 1a). The first phase consisted of a continuous recognition task conducted during fMRI scanning. The second phase consisted of a final memory test conducted outside of the scanner. During the continuous recognition phase, participants viewed 9209–10,000 natural scene images across 30–40 fMRI sessions and indicated whether or not each image had previously been encountered at any point in the experiment (Fig. 1b). Each image was presented up to three times with these exposures pseudo-randomly distributed across the entire experiment

(Fig. 1d). At least two days after completion of the last session of the continuous recognition phase, participants completed a final memory test on a subset of images (Fig. 1c). Each trial of the final memory test began with a recognition memory judgment on a 1–6 confidence scale (1: 'high confidence new', 6: 'high confidence old'). For images judged to be 'old', participants were also prompted to make frequency and temporal memory judgments. For the frequency judgment, participants were asked how many times they had seen the image during the continuous recognition phase (1, 2, 3, or 4 or more). For the temporal memory judgment, which is the primary focus of the present study, participants were instructed to position a marker along a continuous timeline when in the experiment each image was first encountered.

All participants performed above chance on the recognition memory test (Fig. 2a; hit rate greater than false alarm rate: $t_7 = 8.24$, $p < 0.001$, Cohen's $d = 1.56$, 95% confidence interval (CI) = [0.2, 0.36], two-tailed paired-sample $t$-test). Separating the data across three confidence levels (low, medium, and high) revealed that recognition memory accuracy (d') increased with levels of subjective confidence (Fig. 2b; $F_{2,14} = 16.66$, $p < 0.001$, $\eta^2 = 0.70$, one-way repeated-measures ANOVA). Results for the frequency test are reported in Supplementary Fig. 1.

Of critical interest was the accuracy of temporal memory judgments, which required participants to recall the first time each scene was encountered over the course of the up to 10-month experiment. To reduce the effects of non-linearity in temporal memory judgments (e.g., response bias towards the center of the timeline, see "Methods" and Supplementary Fig. 2), we converted both the actual (objective) and the estimated (subjective) temporal positions to ranked positions for further analyses. Based on the ranks, we quantified item-wise temporal memory error by comparing the distance between the actual and estimated ranked positions (Fig. 1e). To determine temporal accuracy across participants, we ran a mixed-effects linear regression model for estimated against actual temporal position with participants as a random effect. Results from this analysis indicated that participants were able to place images in their correct temporal contexts with above-chance accuracy (Fig. 2c; group-level $\beta = 0.302$, $p < 0.001$, 95% CI = [0.24, 0.36]). We further evaluated temporal memory accuracy for each participant using a permutation test (see "Methods"). This analysis revealed that temporal memory performance was above chance for seven out of the eight participants (Fig. 2d; $p$s < 0.01; one participant: $p = 0.083$). The relatively high accuracy of the temporal memory judgments is notable when considering that participants were not informed that they would be tested on temporal memory until after all of the continuous recognition sessions.

### CA1 and entorhinal representational similarity across exposures predicts temporal memory precision

The primary goal of the present study was to investigate whether the similarity (or dissimilarity) of MTL representations across repeated stimulus encounters predicts the accuracy of temporal memory judgments across long timescales. Accordingly, we examined the representational similarity between exposures of each of the images that were subsequently probed in the temporal memory test. Given our a priori interest in MTL structures, we focused on two manually segmented subfields of the hippocampus (CA1 and CA2/3/dentate gyrus, hereafter CA2/3/DG), along with ERC, perirhinal cortex (PRC), and parahippocampal cortex (PHC) (Fig. 3a). For each region of interest (ROI), we correlated the activity patterns between each pair of exposures of the same image (i.e., r(E1, E2), r(E2, E3), and r(E1, E3)). The resulting correlations were then Fisher-transformed prior to statistical testing. As a first step, we averaged across these pairwise correlations to generate a single similarity metric (across exposures) for each image (Fig. 3b). We then compared these similarity metrics for images associated with high versus low temporal memory precision (based on a participant-specific median split). Statistical

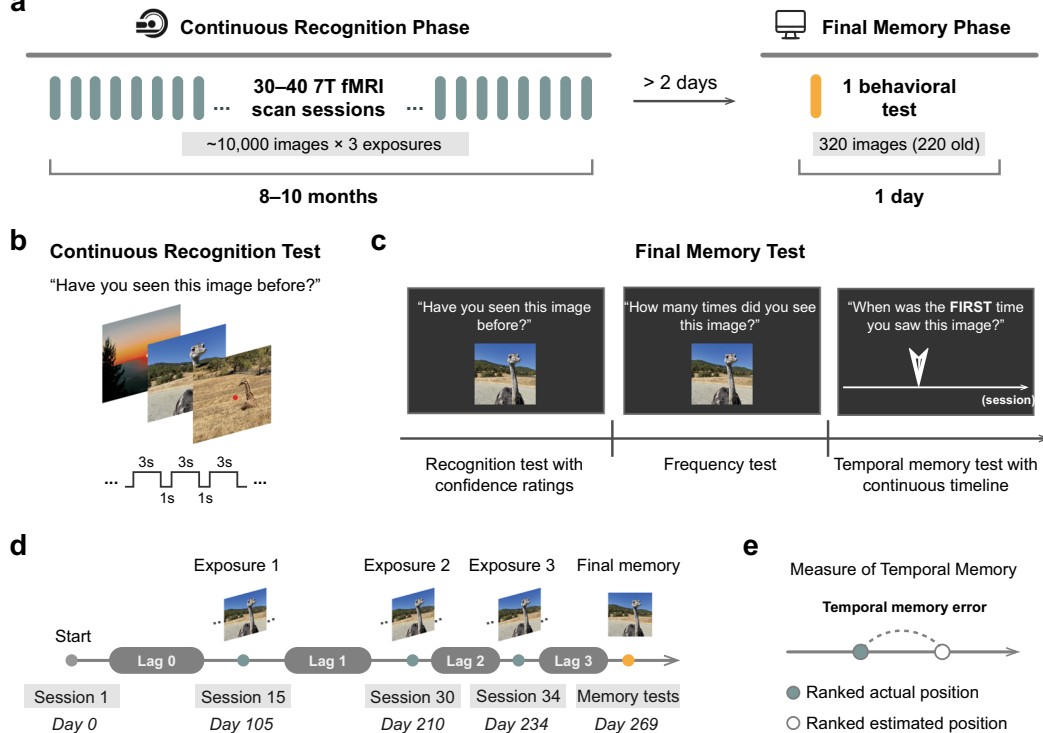

**Fig. 1 | Experimental design. a** Overview of experimental procedures: participants completed two experimental phases. The continuous recognition phase consisted of 30–40 separate fMRI scan sessions distributed across 8–10 months. Across these sessions, thousands of natural scene images were pseudo-randomly presented up to three times. After all of the scan sessions were completed, participants performed a final memory test on a subset of images outside of the scanner on a separate day (2–7 days later). **b** Continuous recognition test. While maintaining central fixation, participants viewed sequences of natural scenes and reported whether each image had been seen at any previous point in the experiment. **c** Final memory test. Each trial of the final memory test began with a recognition memory judgment in which participants made a recognition decision together with a confidence rating from 1–6 (1: 'high confidence new', 6: 'high confidence old'). For each image judged as 'old', a frequency test followed in which participants were asked how many times they had seen the image before (1, 2, 3, or 4 or more). Following that, participants were asked to indicate on a continuous timeline when the image in question was *first encountered* (temporal memory test; note this is a conceptual illustration of the task, see "Methods" and Supplementary Movie 1 for more information). **d** Timeline of an example image. Each old image used in the final memory test was presented three times during the continuous recognition phase and associated with four temporal lags. The first fMRI scan session of the continuous recognition phase for each participant corresponds to Day 0. All temporal lags were quantified in seconds and transformed with the natural logarithm for further analyses. **e** Behavioral measure of temporal memory. Item-wise temporal memory error was quantified as the difference between the ranked actual and ranked estimated temporal positions.

significance of the difference between high and low temporal memory precision was evaluated using a permutation test that shuffles the images' temporal memory identities within each participant. Among the set of MTL ROIs, CA1 and ERC exhibited significantly greater pattern similarity across repeated exposures for high-precision images relative to low-precision images (Fig. 3c; CA1: $p = 0.004$; ERC: $p = 0.004$; permutation tests). The fact that temporal memory precision was associated with greater pattern similarity across exposures in CA1 and ERC is consistent with a context reinstatement account, wherein the original temporal context is reinstated (and strengthened) during subsequent exposures.

We next performed several control analyses. First, because temporal memory precision increased as a function of the session position in which the first exposure occurred (recency effect, see Supplementary Fig. 3), we repeated the analyses for CA1 and ERC while explicitly accounting for temporal lag information (Fig. 1d). Specifically, we ran a mixed-effects logistic regression model that predicted temporal memory precision from pattern similarity across exposures with temporal lags (lag 0–3) included as fixed effects and participant included as a random effect. This analysis confirmed that the relationship between pattern similarity in CA1/ERC and temporal memory precision remained significant when accounting for temporal lag information (Fig. 3d; CA1: $\beta = 2.134$, $p = 0.005$, 95% CI = [0.63, 3.64]; ERC: $\beta = 3.207$, $p = 0.008$, 95% CI = [0.83, 5.58]).

Second, we repeated the foregoing analyses for an early visual cortex ROI (V1) that would be sensitive to low-level visual information but would not be expected to code for temporal context. As expected, V1 pattern similarity across exposures did not differ for high- versus low-precision images (Fig. 3c; $p = 0.25$; permutation test) and was not a predictor of temporal memory precision (Fig. 3d; $p = 0.376$; logistic mixed-effects regression). Likewise, an additional, exploratory whole-brain analysis did not identify any cortical areas outside the MTL for which the relationship between pattern similarity and temporal memory was significant after correction for multiple comparisons (Supplementary Table 1).

Third, and critically, we next tested whether the effects observed in CA1 and ERC were specific to temporal memory. To this end, we repeated the same mixed-effects regression model but now used recognition confidence as the dependent variable instead of temporal precision. Neither CA1 nor ERC exhibited significant relationships between pattern similarity and recognition confidence ($p$s > 0.10). In contrast, pattern similarity was a significant predictor of recognition confidence in PHC (Fig. 3e; $\beta = 0.799$, $p < 0.001$, 95% CI = [0.34, 1.25]). A follow-up control analysis which included recognition confidence together with pattern similarity as fixed effects in a mixed-effects regression model confirmed that pattern similarity in CA1 and ERC predicted temporal memory precision when accounting for recognition confidence (CA1: $\beta = 2.073$, $p = 0.007$, 95% CI = [0.56, 3.58]; ERC:

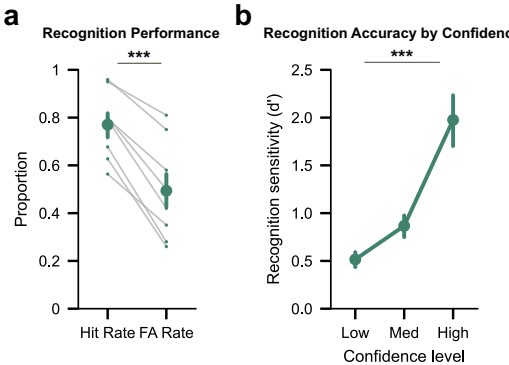

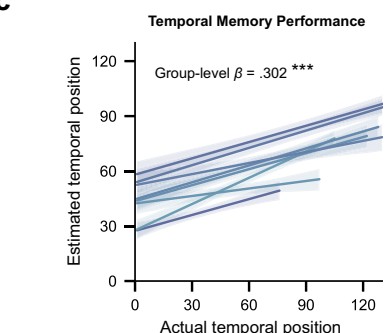

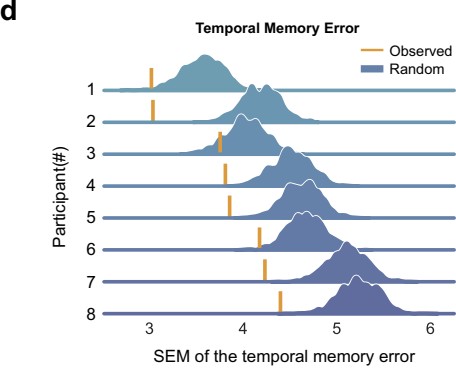

**Fig. 2 | Behavioral results. a** Recognition performance for each participant quantified by hit rate and false alarm (FA) rate. Hit rates were reliably above FA rates (two-tailed paired *t*-test; $t_7 = 8.24$, $p < 0.001$, Cohen's $d = 1.56$, 95% CI = [0.2, 0.36]), indicating above-chance recognition memory. **b** Overall recognition performance (d′) separated by confidence levels. Recognition accuracy increased with subjective confidence levels (one-way repeated-measures ANOVA; $F_{2,14} = 16.66$, $p < 0.001$, $\eta^2 = 0.70$). **c** Correlation between estimated and actual temporal positions. Participants showed above-chance accuracy in temporal memory judgments (group-level $\beta = 0.302$, $p < 0.001$, 95% CI = [0.24, 0.36], linear mixed-effects regression, $n = 8$ independent participants). Each color shaded line indicates a participant. **d** Individual participant's temporal memory performance compared to chance level. Density plots compare the standard error of the mean (SEM) of the observed temporal memory error (yellow line) to the null distribution (blue density; estimated by permuting estimated temporal judgments across images within each participant, $n = 1000$ permutations). Throughout the figure, error bars reflect mean ± s.e.m.; dots or colors denote individual participants ($n = 8$); ***$p < 0.001$. Source data are provided as a Source data file.

$\beta = 3.132$, $p = 0.010$, 95% CI = [0.75, 5.51]). These results help constrain accounts of why pattern similarity in CA1/ERC predicted temporal memory precision. Namely, they argue against the possibility that the relationships between CA1/ERC pattern similarity and temporal memory precision were a secondary consequence of overall memory strength for the images. Rather, pattern similarity across exposures in

CA1 and ERC specifically predicted better memory for when (in time) images were first encountered.

### Similarity between first and second exposures uniquely predicts temporal memory

Having demonstrated that CA1 and ERC pattern similarity across repeated exposures predicts temporal memory for an image's first exposure, we next sought to determine which pair of image exposures was most predictive of temporal memory. From a context reinstatement perspective, similarity between the first exposure (E1) and the second exposure (E2) should be uniquely important because E2 provides the first opportunity to reinstate the temporal context from E1. To test this, we first compared pattern similarity for high- and low-precision images for each pair of image exposures (E1-E2, E2-E3, and E1-E3). Statistical significance of the difference between high- and low-precision images for each exposure pair was computed a permutation analysis in which, for each participant and exposure pair, we randomly shuffled the images' temporal memory precision labels. For both CA1 and ERC, E1-E2 similarity was significantly greater for high- than low-precision images (Fig. 4a; CA1: $p = 0.015$; ERC: $p = 0.007$; permutation tests). However, both regions also exhibited similar effects for E2-E3 similarity (Fig. 4a; CA1: $p = 0.025$; ERC: $p = 0.036$, permutation tests). Neither region exhibited a significant effect for E1-E3 similarity (Fig. 4a; *p*s > 0.28).

To further explore this pattern of results, we performed three follow-up sets of analyses. First, in order to control for potential temporal lag effects (Supplementary Fig. 3), we ran a mixed-effects logistic regression model that predicted temporal memory from pattern similarity of each exposure pair (E1-E2, E2-E3, and E1-E3 as separate dependent variables in one regression model) while including lag information. For both CA1 and ERC, E1-E2 similarity significantly predicted temporal memory (Fig. 4c; CA1: $\beta = 1.048$, $p = 0.014$, 95% CI = [0.21, 1.88]; ERC: $\beta = 1.565$, $p = 0.022$, 95% CI = [0.22, 2.91]). Effects were marginally significant for E2-E3 similarity (*p*s < 0.10), and not significant for E1-E3 similarity (*p*s > 0.68).

Second, in order to more directly assess whether E1-E2 similarity contained predictive power above and beyond that of other exposure pairs, we compared the performance of several models that did or did not include various exposure pairs. That is, we tested whether model performance was significantly improved when E1-E2 similarity was added to models that only included E2-E3 and E1-E3 similarity. For both CA1 and ERC, adding E1-E2 as a predictor significantly improved the model's performance (CA1: $\chi^2 = 6.147$, $p = 0.013$; ERC: $\chi^2 = 5.315$, $p = 0.021$). In contrast, adding E2-E3 and E1-E3 similarity as predictors to models with just E1-E2 similarity did not improve the model's performance (*p*s > 0.15). These results established that E1-E2 similarity was uniquely important for subsequent temporal memory judgments, as would be predicted by a context reinstatement account.

Third, to further establish whether E1-E2 similarity was uniquely important for temporal memory, we tested for relationships between different exposure pairs and recognition memory confidence. Interestingly, although PHC pattern similarity across exposures was highly predictive of subsequent recognition memory confidence (Fig. 3e), this effect was not driven by E1-E2 similarity (Fig. 4b; $p = 0.482$, permutation test; Fig. 4d; $p = 0.225$; linear mixed-effects regression). Instead, E1-E3 similarity in PHC significantly predicted recognition confidence (Fig. 4b; $p = 0.002$; Fig. 4d; $\beta = 0.473$, $p = 0.018$, 95% CI = [0.08, 0.86]). Along with the results described above, these findings provide a qualitative dissociation between the predictors of temporal memory versus recognition memory. That is, whereas pattern similarity between the first and second exposure of an image was uniquely important for remembering *when* the image was first encountered, it was relatively less important for recognizing *whether* the image was previously encountered.

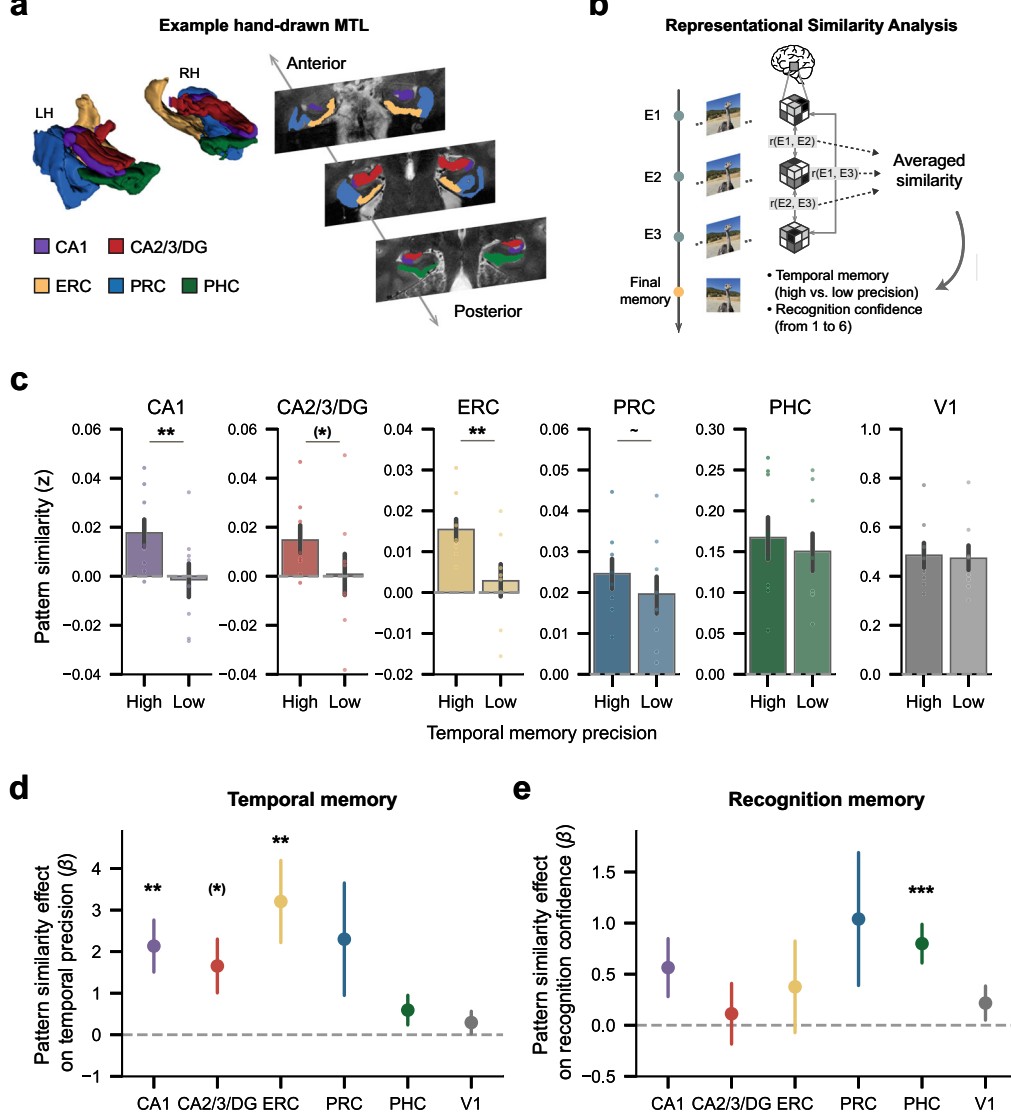

**Fig. 3 | CA1 and entorhinal representational similarity predicted temporal memory precision, but not recognition confidence. a** Manually drawn ROIs for MTL subregions of an example participant: CA1 (purple), CA2/3/DG (red), ERC (yellow), PRC (blue), and PHC (green). LH/RH: left/right hemisphere. **b** Schematic depiction of representational similarity analysis. **c** Pattern similarity difference between high- and low-precision images (median split) across MTL subregions and a control early visual region (V1). CA1 and ERC showed greater pattern similarity across exposures for high-precision images relative to low-precision images (CA1: $p = 0.004$; ERC: $p = 0.004$; one-sided permutation tests, $n = 1000$). CA2/3/DG showed similar effect but did not survive correction for multiple comparisons ($p(uncorrected) = 0.023$). **d** Relationship between pattern similarity across exposures and temporal memory precision. Pattern similarity across repeated exposures in CA1 and ERC predicted temporal memory precision (high vs. low) while

accounting for temporal lag information (CA1: $\beta = 2.134$, $p = 0.005$, 95% CI = [0.63, 3.64]; ERC: $\beta = 3.207$, $p = 0.008$, 95% CI = [0.83, 5.58]; logistic mixed-effects regression, $n = 8$ independent participants). A similar effect was also observed in CA2/3/DG ($p(uncorrected) = 0.037$), but did not survive correction for multiple comparisons. **e** Relationship between pattern similarity across exposures and recognition confidence. Pattern similarity across repeated exposures in PHC predicted recognition confidence while accounting for temporal lag information ($\beta = 0.799$, $p < 0.001$, 95% CI = [0.34, 1.25]; liner mixed-effects regression). Throughout the figure, error bars reflect mean ± s.e.m.; dots denote independent participants ($n = 8$); ~$p < 0.10$; *$p < 0.05$; **$p < 0.01$; ***$p < 0.001$. Parentheses indicate ROIs that did not survive multiple comparison correction. Source data are provided as a Source data file.

## CA1 and ERC predict temporal memory via image-specific representations

While all of the preceding representational similarity analyses were performed by correlating activity patterns across repeated exposures of the same stimulus (i.e., image-specific correlations), these analyses do not guarantee that the information that predicted temporal memory precision was specific to individual images. Namely, it is possible that temporal memory precision benefited from generic memory processes or attentional states that generalized across images (e.g., states optimized for memory encoding[31]). While this possibility would still support a role for CA1 and ERC in encoding temporal information,

a temporal context reinstatement account fundamentally predicts reinstatement of the specific temporal context in which an image was encoded.

To assess whether temporal memory was predicted by image-specific pattern similarity, we conducted two additional analyses (restricted to E1-E2 similarity). First, for all of the images tested in the temporal memory test, we permuted the E1-E2 mappings by shuffling images' E2 within each participant. We then calculated the resulting E1-E2 pattern similarity scores and a corresponding distribution of beta values reflecting the relationships with temporal memory (see "Methods" for details). Critically, for both CA1 and ERC, the

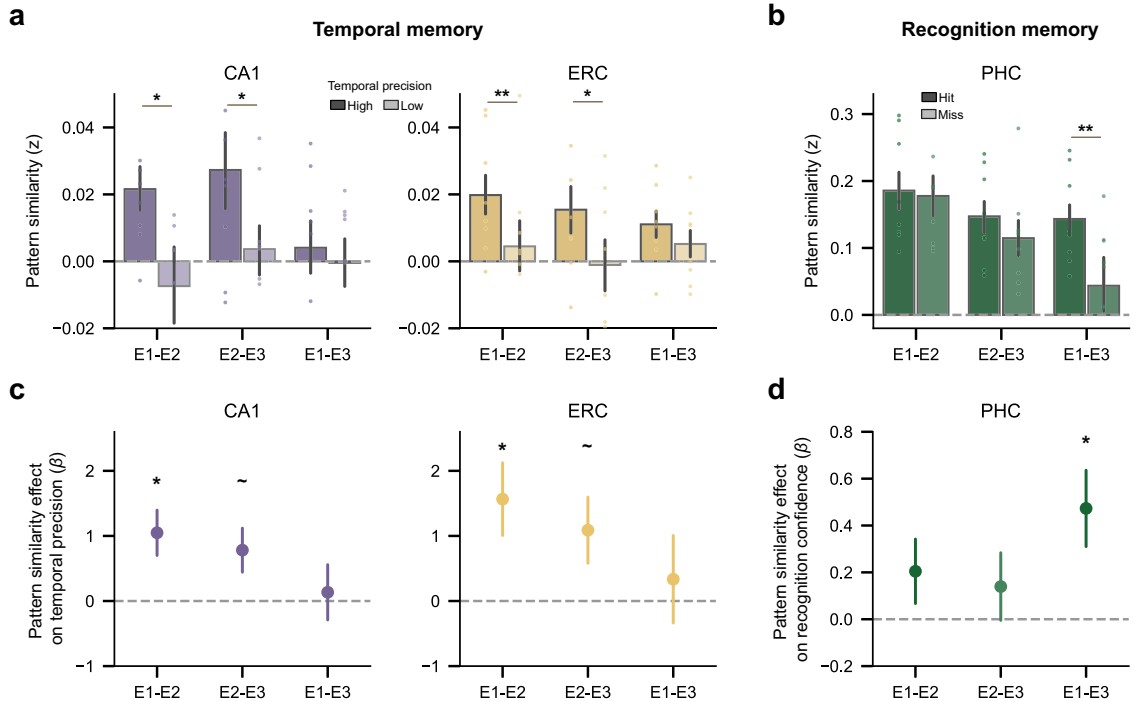

**Fig. 4 | Pattern similarity between the first and second exposure in CA1 and ERC was uniquely important for temporal memory. a** CA1/ERC pattern similarity between high- and low-precision images for each pair of image exposures. CA1 and ERC showed greater pattern similarity for high-precision images relative to low-precision images in E1-E2 (CA1: $p = 0.015$; ERC: $p = 0.007$; permutation test, $n = 1000$) and E2-E2 (CA1: $p = 0.025$; ERC: $p = 0.036$; permutation test). **b** PHC pattern similarity between hits and misses in recognition memory for each pair of image exposures. PHC showed greater pattern similarity for hits relative to misses in E1-E3 ($p = 0.002$; permutation test, $n = 1000$). **c** Relationship between pattern similarity for each pair of image exposures in CA1/ERC and temporal memory precision while accounting for temporal lag information. For both CA1 and ERC, E1-E2 pattern similarity was significantly predictive of temporal memory precision (CA1: $\beta = 1.048$, $p = 0.014$, 95% CI = [0.21, 1.88]; ERC: $\beta = 1.565$, $p = 0.022$, 95% CI = [0.22, 2.91]; logistic mixed-effects regression, $n = 8$ independent participants). **d** Relationship between pattern similarity for each pair of image exposures in PHC and recognition confidence while accounting for temporal lag information. Recognition confidence was predicted by E1-E3 pattern similarity in PHC ($\beta = 0.473$, $p = 0.018$, 95% CI = [0.08, 0.86]; linear mixed-effects regression). Error bars reflect mean ± s.e.m.; dots denote independent participants ($n = 8$); -$p < 0.10$; *$p < 0.05$; **$p < 0.01$. Source data are provided as a Source data file.

relationship between 'intact' E1-E2 similarity and temporal memory was significantly stronger (higher beta values) than the permuted values (Fig. 5a; CA1: $p = 0.019$; ERC: $p = 0.025$). These data provide important evidence that temporal memory precision was predicted by image-specific pattern similarity in CA1 and ERC.

As a follow-up to the preceding analysis, we ran a final analysis to address whether apparent image-specific effects might be due to general memory states and/or differences in coarse temporal context information (i.e., session effects). Thus, for each image included in the temporal memory test (a 'target'), we identified control images ('foils') such that the targets and foils shared the same E1 session number, but not scanning run (to avoid potential contamination from auto-correlation in the fMRI data), and the same E2 session number (but not run; Fig. 5b). To match recognition memory with targets, foils were only included in this analysis if they were correctly rejected at E1 and successfully recognized at E2 and E3 (see "Methods" for details). This allowed us to compute similarity between target E1 and target E2 (target similarity) and target E1 and foils E2 (foil similarity). The difference between these measures (target similarity − foil similarity) was then used as a predictor of temporal memory precision. Indeed, this similarity difference score significantly predicted temporal memory precision for both CA1 (Fig. 5c; $\beta = 0.864$, $p = 0.033$, 95% CI = [0.07, 1.66]) and ERC (Fig. 5c; $\beta = 1.308$, $p = 0.047$, 95% CI = [−0.02, 2.60]). These findings lend further support to the idea that temporal memory precision was related to image-specific pattern similarity measures and specifically argue against potential confounds due to generic memory-related processes or session effects. The fact that these effects held when carefully controlling for session effects is notable because it

provides evidence against the possibility that pattern similarity only captured coarse-level temporal context (session information). Rather, to the extent that the pattern similarity measure captured temporal context information, these findings suggest a relatively 'local' temporal context representation that differentiated between images within the same session (day).

## Discussion

The ability to remember when events occurred in time is fundamental to human experience. However, retaining precise temporal memories is complicated by the fact that real-world episodic memories span long timescales (days, weeks, months and beyond) and by the fact that events may re-occur in multiple contexts over those long timescales (e.g., a movie you have viewed several times over the past year). To date, there is remarkably little evidence characterizing how the human brain preserves temporal memories in the face of these challenges. Here, we show that when events re-occur over long timescales (at lags up to several months), the re-expression of distributed, event-specific activity patterns in CA1 and ERC preserves memory for the original temporal context of an event (i.e., memory for when an event first occurred). These findings are consistent with and bridge between prior human and rodent studies implicating CA1 and ERC in temporal processing and temporal memory. However, our findings also go beyond existing evidence by providing a mechanistic account of how CA1 and ERC preserve temporal memories and demonstrating these relationships at uniquely long timescales.

While there is a rich history characterizing temporal memory in human behavioral and neuroimaging studies[32,33], it is striking how few

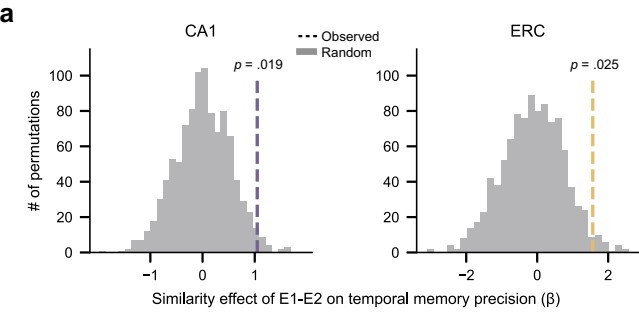

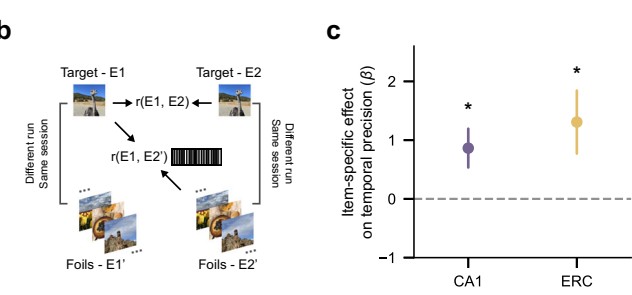

**Fig. 5 | Representational image-specificity analyses. a** Intact compared to permuted similarity effect. E1-E2 pattern similarity compared to permuted similarity exhibited a stronger effect on temporal memory precision in both CA1 and ERC (CA1: $p = 0.019$; ERC: $p = 0.025$; permutation tests, $n = 1000$). **b** Schematic illustration showing how target similarity and foil similarity were computed for an example image (see "Methods" for details). **c** Relationship between image-specific pattern similarity (target similarity − foil similarity) in CA1/ERC and temporal memory precision. Image-specific pattern similarity was significantly predictive of temporal memory precision in both CA1 and ERC (CA1: $\beta = 0.864$, $p = 0.033$, 95% CI = [0.07, 1.66]; ERC: $\beta = 1.308$, $p = 0.047$, 95% CI = [−0.02, 2.60]; logistic mixed-effects regression, $n = 8$ independent participants). Error bars reflect mean ± s.e.m.; -$p < 0.10$; *$p < 0.05$. Source data are provided as a Source data file.

of these studies have considered temporal memory across timescales that exceed a single experimental session. Indeed, our approach of testing temporal memory for images that were distributed across dozens of experimental sessions/scans spanning 8–10 months is unprecedented. Considering that the overwhelming majority of real-world episodic memories span days, weeks, months and years, it is imperative to understand the neural mechanisms that support temporal memory at these timescales. Although it is intuitively obvious that humans can and do retain temporal memories over long timescales, it is nonetheless remarkable that participants in the current study were generally successful at recalling the initial temporal context for images presented at the final memory test given that (a) these images were drawn from a pool of tens of thousands of images, (b) the delay between the initial exposure and the final memory test ranged from days to almost a year, and (c) each image was presented in multiple temporal contexts, creating potential interference. Thus, by simulating the challenges that are inherent to real-world temporal memory, our experimental paradigm provides a unique opportunity to characterize the underlying neural mechanisms.

By leveraging representation-based analyses to track patterns of activity across repeated stimulus exposures and distinct temporal contexts, we were able to gain critical insight into the mechanisms through which CA1 and ERC contribute to temporal memory. In particular, our findings strongly align with a context reinstatement account. According to temporal context models[20,27], context representations—reflected in distributed patterns of neural activity—gradually change over time and are reinstated when a stimulus is subsequently remembered[34–40]. This temporal context reinstatement is thought to directly support the ability to remember—or infer—when in time stimuli were previously encountered[29]. From this perspective,

our finding that greater pattern similarity across exposures preserved memory for an event's original temporal context can be explained in terms of the original context representation (elicited during E1) being reinstated and strengthened during subsequent exposures (E2, E3)[41] and ultimately guiding temporal memory judgments at the end of the experiment. In fact, this account also readily explains our finding that similarity between the first and second exposure (E1, E2) was uniquely important for temporal memory. Namely, E2 represented the first potential 'reminder' of E1's temporal context. Interestingly, although we tested for re-expression of E1's activity patterns by explicitly re-exposing participants to the same stimulus multiple times (E2, E3), our findings likely generalize to situations where stimuli are not explicitly re-exposed (or re-encountered). Indeed, human neuroimaging studies (unrelated to temporal memory) have found that offline, spontaneous reinstatement of episodic memories not only occurs, but it strengthens memories in much the same way that online, cued reinstatement does[42].

While a context reinstatement account makes a clear prediction that better memory for the original temporal context should be associated with greater representational similarity across exposures, it is notable that a memory interference account[25] suggests an entirely opposite prediction: that temporal memory would benefit from greater contextual distinctiveness across exposures (i.e., less similarity). More specifically, greater contextual distinctiveness would putatively be expected to reduce interference between the various temporal contexts in which an event occurred (E1, E2, E3). That said, there are several examples in the memory interference literature where reinstatement of prior experiences during new learning can, in fact, protect memories from interference[43,44]. Moreover, it is important to note that a context reinstatement account for CA1 and ERC does not exclude the possibility that other MTL regions (e.g., CA3) might simultaneously contribute to temporal memory by emphasizing differences between temporal contexts[45–48].

The fact that we specifically identified CA1 and ERC as being important for temporal memory at long timescales—and the implication that these regions supported temporal context reinstatement—is striking in light of accumulating evidence documenting time sensitive cells within rodent CA1 and ERC[12–14]. It has been speculated that ensembles of these cells allow for the coding of gradually-drifting temporal context representations which become bound to individual events[19] and reinstated when events are remembered[20,27]. Importantly, although individual cells may only operate across very short timescales (seconds), ensemble representations can track temporal information over much longer timescales—from minutes to days[18,49]. Here, we were not able to directly measure or identify time cells because, even with the relatively high spatial resolution of the current fMRI data, each voxel likely pooled across tens or hundreds of thousands of cells. However, the representation-based analyses we employed are potentially well-suited to capturing gradual changes in ensemble-level context representations[26,35,36,50–53]. In contrast, although several prior studies of human memory have also implicated CA1 and ERC in memory for when events occurred[21,22], most of these studies have not employed representation-based analyses and, therefore, are not amenable to testing or capturing temporal context representations. Thus, our approach and findings uniquely bridge between evidence of time cells in rodents, theoretical models of temporal context, and prior studies of temporal memory in humans.

An additional essential consideration in understanding neural mechanisms that specifically relate to temporal memory is to establish that any apparent effects related to temporal memory were not derivative from more general effects of memory strength. Specifically, as memories decay over time, temporal judgments could potentially be inferred from the strength of memories themselves[54–56]. This is of particular concern given the very long timescales involved in the current study. However, several theoretical perspectives propose that

memory for time is dissociable from memory strength[33,57,58]. Here, our final memory test separately measured recognition confidence (a proxy for overall memory strength) and temporal memory, allowing us to conduct several targeted analyses aimed at teasing apart these two expressions of memory. First, we found that the relationships between CA1/ERC and temporal memory precision remained significant in a regression model that included recognition confidence as a covariate. Second, consistent with prior arguments that distinct MTL subregions are involved in 'item-based' versus 'context-based' memory[59], we found that pattern similarity measures in PHC predicted recognition confidence but not temporal memory, whereas pattern similarity measures in CA1 and ERC predicted temporal memory but not recognition confidence. Finally, when considering pattern similarity across specific pairs of image exposures, temporal memory (defined here as memory for when the first exposure occurred) was best predicted by pattern similarity between the first and second exposures, consistent with a context reinstatement account. In contrast, recognition confidence was best predicted by pattern similarity between the first and third exposures, potentially indicating that the last (third) exposure was relatively more influential to memory strength (also see Supplementary Fig. 3). Together, these data points provide important, converging evidence that temporal memory judgments in the current study were not derived from the overall memory strength. More generally, our findings reinforce theoretical accounts that emphasize the distinction between memory for 'when' an event occurred versus 'whether' an event occurred[5,6,60,61].

In conclusion, here we show that memory for the temporal context in which an event initially occurred is preserved via the re-expression of activity patterns in human CA1 and ERC. Critically, we show that these dynamics operate across—and support memory at—long timescales (from days to months). These findings complement yet significantly advance existing evidence from rodents and humans implicating the hippocampal-entorhinal system in representing and remembering time. In particular, our findings suggest that distributed patterns of activity in CA1 and ERC encode and reinstate temporal context information, thereby preserving memory for when events occurred.

## Methods

### Participants
Eight participants took part in the study (two self-identified males, six self-identified females; age range: 19–32). All participants were right-handed with no known cognitive deficits nor color blindness and with normal or corrected-to-normal vision. Participants were naïve to the experimental manipulation and were not involved in the design nor planning of the study. Informed written consent was obtained from all participants before the start of the study, and the experimental protocol was approved by the University of Minnesota Institutional Review Board.

### Design and procedure
Data used in this study were collected as part of the Natural Scenes Dataset (NSD; http://naturalscenesdataset.org), and included two parts: a continuous recognition phase conducted in the fMRI scanner and a behavioral final memory phase (Fig. 1a).

**Continuous recognition phase.** A detailed description of the continuous recognition phase has been reported in a previous publication[30]. Briefly, for each participant, the continuous recognition phase was split across 40 scan sessions in which 10,000 distinct color natural scenes would be presented three times spaced pseudorandomly over the course of all scan sessions using Psychophysics Toolbox 3.0.14. Each scan session consisted of 12 runs (750 trials). Distributions of image presentations were controlled such that both short-term and long-term re-exposures were probed (see Stimuli

section below). Four of the participants completed the full set of 40 NSD scan sessions. Due to constraints on participant and scanner availability, each of the other four participants completed 30–32 scan sessions. In these collected data, each participant viewed 9209–10,000 distinct images and participated in 22,500–30,000 trials. Each trial lasted 4 s, consisting of the presentation of an image for 3 s and a following 1-s gap. Participants were instructed to perform a continuous recognition task in which they reported whether the current image had been seen at any previous point in the experiment.

**Final memory phase.** At least two days (range: 2–7 days) after completion of the continuous recognition phase, a final memory test was administered outside of the scanner. Participants were not informed about the final memory test in advance. During the final memory phase, participants viewed a subset of old images (220 per participant) from the continuous recognition phase randomly intermixed with novel images (100 per participant) and completed different types of memory probes. The final memory phase consisted of 320 trials, with up to three judgments per trial. Each trial began with a recognition test in which participants performed an old or new judgment with a confidence rating on a scale of 1 to 6 (1: 'high confidence new', 2: 'medium confidence new', 3: 'low confidence new', 4: 'low confidence old', 5: 'medium confidence old', 6: 'high confidence old'). For images judged as "old", a frequency test followed in which participants were asked to indicate how many times they had seen each image (1, 2, 3, or 4 or more times). Following the frequency test, participants performed a temporal memory test using a timeline. In this test, participants were asked to indicate, on a continuous timeline with tick marks to represent each session, when in the experiment they thought each image was first encountered (Fig. 1c, right). The length and labels of the timeline vary across participants, depending on how many sessions they completed in the continuous recognition phase. Participants were encouraged to use the full length of the scale, with the left endpoint representing the beginning of the continuous recognition phase and the right endpoint representing the end. Participants used a cone to mark the temporal location on the line and were instructed to indicate their confidence in response via adjusting the size of the cone, with smaller cones representing higher confidence and bigger cones representing lower confidence (see Supplementary Movie 1 for depiction of example trials). Given the primary focus of the present study concerns temporal memory precision, we only analyzed the estimates of temporal location as illustrated in Fig. 1c. All tests in the final memory phase were self-paced with a timeout of 30 s.

### Stimuli
All images used in this study were taken from the Microsoft Common Objects in Context (COCO) database[62].

**Continuous recognition phase.** For the continuous recognition phase, a total of 73,000 images were prepared with the intention that each participant would view 10,000 distinct images (9000 unique images and 1000 shared images across participants) three times each over the course of 40 scan sessions. To prevent the recognition task from becoming too difficult (and risking loss of morale), each image was randomly placed three times on a circle according to a probability distribution created by mixing a relatively narrow von Mises distribution and a uniform distribution. Across all scan sessions, the mean number of distinct images shown once, twice, and all three times within a typical session is 437, 106, and 34, respectively.

**Final memory phase.** The final memory phase included a total of 320 images for each participant: 220 old images and 100 novel images. Old images were selected from the continuous recognition phase that participants completed during fMRI scanning with each image having been presented three times during the continuous recognition phase.

Novel images were from the COCO dataset but were not presented during the continuous recognition phase.

The 220 old images included in the final memory test were comprised of two sets of images selected according to different criteria. The first set comprised 120 images that were selected based on behavioral accuracy and the distribution of exposures during the continuous recognition phase. Specifically, these images were selected based on the following three criteria: (1) Each of the 120 images was associated with a correct behavioral response at each exposure during the continuous recognition phase—i.e., correct rejection (E1), hit (E2), and hit (E3). (2) To promote overall temporal memory performance, approximately half of the selected images were associated with a first exposure (E1) that occurred during the last eight scan sessions (for one participant, this was adjusted to the last ten scan sessions in order to have enough trials given their performance in the continuous recognition phase); the other half of the selected images were associated with a first exposure that occurred from the rest of the scan sessions (i.e., earlier sessions). (3) For each half of the images, there was an additional constraint on the spacing between exposures, with one-third of the images having all three exposures within one scan session, one-third with the last two exposures in the same session, and the rest either with the first two exposures in the same session or with all three exposures across different sessions.

The second set of 100 old images included during the final memory test were selected to maximally span semantic space (see the NSD data paper[30] for details), without consideration of behavioral accuracy and distribution of exposures. Briefly, this was done by computing shifted inverse frequency sentence embeddings for the sentence captions, and using a greedy approach to determine the subset of 100 images that maximize the average distance between each image's embedding and its closest neighbor. The motivation for sampling semantic space was unrelated to the goals of the current manuscript. However, in order to increase statistical power, we opted to include images from the second set in our analyses but only for images that were associated with correct behavioral responses for each of the three exposures in the continuous recognition phase (as was the case for the first set). This resulted in a total of 143-170 images that were used for analyses for each participant (see Supplementary Fig. 4 for the distribution of images' first exposures across participants).

## MRI data acquisition and preprocessing

The imaging data was collected as part of the NSD at the Center for Magnetic Resonance Research at the University of Minnesota. In brief, functional data and a few additional anatomical measures were collected using a 7T Siemens Magnetom passively-shielded scanner with a single-channel-transmit, 32-channel-receive RF head coil (Nova Medical, Wilmington, MA). Functional data was acquired using whole-brain gradient-echo echo-planar imaging (EPI) at 1.8-mm resolution and 1.6-s repetition time. In addition to the EPI scans, for the purposes of hippocampal segmentation, a high-resolution $T_2$-weighted scan was acquired during one of the 7T scan sessions. $T_1$- and $T_2$-weighted structural scans were collected using a combination of a 3T Siemens Prisma scanner and a standard Siemens 32-channel RF head coil.

Functional data were pre-processed by performing one temporal resampling to correct for slice time differences and one spatial resampling to correct for head motion within and across scan sessions, EPI distortion and gradient non-linearities. Informed by the original data paper[30], the current study used the upsampled 1.0-mm high-resolution preparation of the NSD data in order to optimally partition the functional data into regions of interest defined using high-resolution anatomical images.

Parameter estimates (beta weights) reflecting fMRI response amplitudes evoked by each trial were estimated using a general linear model (GLM) approach as described in the NSD data paper[30]. Notably,

our approach (which corresponds to "beta version 2" in the NSD data paper) involved generating voxel-specific hemodynamic response functions (HRFs). Briefly, the pre-processed time-series data were fitted multiple times with a single-trial GLM, each time using a different HRF from a library of HRFs. For each voxel, the HRF that provided the best fit to the data was identified and single-trial betas were then generated using that HRF. Betas were then converted to units of percent BOLD signal change by dividing amplitudes by the mean signal intensity observed at each voxel and multiplying by 100. Our decision to use voxel-specific HRFs was made a priori and was motivated by evidence from the original data paper that it leads to fewer artifacts.

## Regions of interest (ROIs)

The medial temporal lobe (MTL) ROIs were manually drawn on the high-resolution $T_2$ images obtained for each participant, following a 7T protocol for segmentation of MTL subregions[63]. Labels were defined on the raw high-resolution $T_2$ volume, and were mapped via an affine transformation to subject-native anatomical space. The MTL ROIs included bilateral CA1, CA2/3/dentate gyrus, entorhinal cortex (ERC), perirhinal cortex (PRC), and parahippocampal cortex (PHC). Example MTL ROIs from one participant were depicted in Fig. 3a. We also included the primary visual cortex (V1) as a control region. The bilateral V1 ROI was manually drawn on cortical surfaces based on results of a population receptive field experiment from the NSD, and were then mapped to volumetric format. Cortical ROIs for the whole-brain parcel level analysis were defined by a multi-modal cortical parcellation from the Human Connectome Project[64].

## Behavioral data analyses

Overall performance for the temporal memory test was quantified by regressing each participant's subjective estimate of when an image was first encountered against the actual (objective) time (Fig. 2c). Note that there is a general response bias among participants toward the center of the timeline ("raw estimated position", see Supplementary Fig. 2). To account for this response bias and potential non-linearity, the estimated and actual temporal positions used in all analyses in the current paper were converted to ranks according to each individual's marked positions on the timeline and the actual temporal positions in the continuous recognition phase, respectively. To quantify item-wise temporal memory error, we calculated the absolute difference between the ranked estimated temporal position and the ranked actual position (Fig. 1e). To test whether each participant had above-chance temporal memory performance, we compared the observed temporal memory error against a null distribution of permutations (1000 iterations), in which the subjective estimates were randomly shuffled across trials for each participant and the temporal memory error was recomputed for each iteration. Due to the non-normal distribution of temporal error (absolute value of the difference of two rank distributions, see Supplementary Fig. 5), we divided temporal memory trials into 'high-precision' and 'low-precision' groups by performing a median split of temporal error for each participant. This was an a priori decision (i.e., made before conducting any fMRI analyses).

To control for temporal lag information and test for relationships between lag and subsequent memory performance (Supplementary Fig. 3), as illustrated in Fig. 1d, four temporal lags were calculated for each image: the lag between the beginning of the continuous recognition phase and the first exposure (lag 0), the lag between the first and second exposure (lag 1), the lag between the second and third exposure (lag 2), and the lag between the third exposure and the final memory phase (lag 3). The first scan session of the continuous recognition phase for each participant corresponds to Day 0. Because memory is observed to abide by an exponential rule rather than linear time[65], all temporal lags were quantified by expressing time intervals in

seconds and transforming these intervals with the natural logarithm. Lag effects were then tested using mixed-effects regression models with either recognition confidence or temporal memory precision as a dependent variable and with each temporal lag as a separate predictor.

### Representational similarity analyses

Representational similarity analyses were conducted on functional data (single-trial betas) from the continuous recognition phase, and were performed by assessing patterns of neural activity across voxels within each ROI evoked during single trials. Pattern similarity of all possible exposure pairings (Fig. 3b; r(E1, E2), r(E2, E3), and r(E1, E3)) for each image was computed using Pearson correlation. The resulting correlation coefficients were then Fisher-transformed for further analyses. To avoid potential contamination of similarity from scanner-induced autocorrelation of signals, only correlations between image exposures that occurred across runs were considered (range of the trials excluded for each participant: 12–35).

### Image-specificity analyses

We used two approaches to assess image-specificity in CA1 and entorhinal representations that predicted temporal memory.

**Intact versus shuffled pattern similarity analysis.** Our first analysis tested whether temporal memory precision was predicted by image-specific pattern similarity (restricted to E1-E2 similarity) in CA1 and ERC using images tested in the temporal memory test (which were a subset of the full image set). Specifically, we randomly shuffled the E1-E2 mappings within each participant, such that each image's E1 was paired with a different image's E2. We then computed the pattern similarity of these shuffled exposure pairs and the new corresponding temporal lags. The shuffled E1-E2 pattern similarity scores and temporal lag information were then submitted to a mixed-effects logistic regression model predicting temporal memory precision. This procedure was performed 1,000 times, resulting in a null distribution of pattern similarity effects (betas values) for each ROI.

**Target versus foil pattern similarity analysis.** Our second approach examined whether pattern similarity effects observed in CA1 and ERC were specific to individual images or were driven by general memory-related processes that could be shared across different images and/or differences in coarse temporal information (i.e., session effects). To do this, for each image included in the temporal memory test (a 'target'), we identified control images ('foils') according to two criteria: (1) targets and foils shared the same E1/E2 session number, but not run number, respectively (Fig. 5a); (2) to control for generic memory states (recognition memory performance at each encounter), foils had to receive the same memory judgments as targets (i.e., to be responded correctly all three times), which were correctly rejected at E1 and hit at E2 and E3. We then computed pattern similarity between target E1 and target E2 ('target similarity') and target E1 and foils E2 ('foil similarity'). This selection procedure resulted in different numbers of foils for each target image. For images with two or more foils, we used the median value of those foil similarity scores. To index the extent to which pattern similarity captures image-specific representations, foil similarity was subtracted from target similarity for each image (target similarity – foil similarity). This difference score between target and foil similarity was then submitted to a mixed-effects logistic regression model as a predictor of temporal memory precision, where a significant positive relationship would indicate that the pattern similarity that predicted temporal memory precision was driven by image-specific representations.

### Statistical analyses

Behavioral and fMRI data were analyzed using a combination of permutation tests, paired $t$ tests, repeated-measures ANOVA, and mixed-effects regression models. Trial-level relationships between similarity measures and final memory performance were tested with mixed-effects linear/logistic regression models (for recognition confidence and temporal memory precision, respectively). For all permutation analyses, we used 1000 permutations and assessed significance by computing the proportion of values in the null distribution that were higher/lower than the observed values. All $t$ tests were two-tailed. For mixed-effects regression models, we used the participant as a random effect and other variables as fixed effects. A threshold of $p < 0.05$ was used to establish statistical significance for all analyses. fMRI analyses were corrected for multiple comparisons with Bonferroni corrections when applicable. Only ROIs that survived correction are reported except where otherwise noted.

### Reporting summary

Further information on research design is available in the Nature Portfolio Reporting Summary linked to this article.

## Data availability

The NSD dataset is freely available at http://naturalscenesdataset.org. Images used for NSD were taken from the Common Objects in Context database (https://cocodataset.org). Source data are provided with this paper.

## Code availability

Code for main analyses can be found at: https://github.com/futingzou/nsdFinalMem.

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

## Acknowledgements

This work is dedicated to Sarah DuBrow, who passed away before manuscript completion and without whom none of this would have happened. We thank Vishnu P. Murty for feedback on previous versions of the manuscript, and members of the DuBrow lab and the Hutchinson lab for helpful discussions. This work was supported by Sloan Research Fellowship FG-2020-13455 (S.D.) and NIH R01 NS089729 (B.A.K.). Collection of the NSD dataset was supported by NSF IIS-1822683 (K.K.) and NSF IIS-1822929 (T.N.).

## Author contributions

F.Z., I.C., J.B.H., and S.D. conceived and designed the final memory test. K.K. and T.N. conceived and designed the NSD main experiment. G.W. performed manual segmentations of the medial temporal lobe. E.J.A. and Y.W. collected the data. F.Z. analyzed the data. F.Z., B.A.K., J.B.H., and S.D. wrote the paper.

## Competing interests

The authors declare no competing interests.
