## [Peer Review File · Nature Communications]

Re-expression of CA1 and entorhinal activity patterns preserves temporal context memory at long timescalesReviewer #1 (Remarks to the Author):

Zou and colleagues use high-resolution fMRI to examine the reactivation of brain activity patterns related to temporal context memory. This study is unique and innovative in its approach, as it leverages a substantial data set where the same participants were scanned in 30-40 MRI scan sessions each, performing a continuous recognition task. In each session, participants view a series of images, indicating for each image whether they have seen it previously in the experiment. They are tested at the end of the set of sessions using a combination of recognition, frequency, and temporal judgments. This allows the authors to examine temporal order memory on a time-scale previously unheard of in human neuroimaging. The work identifies regions in medial temporal lobe as containing information relevant to the precision of one's memory for past time.

Several key phenomena are demonstrated by the research team, which make this an important contribution to our understanding of the brain circuits important for humans' ability to remember the temporal structure of their past experience. They demonstrate that image-specific brain activity is present in CA1 and ERC, and that increased pattern similarity is associated with better memory for when the item was first encountered in the 40-session sequence (during the final test). This is interpreted as evidence that the participant is reactivating a temporal context representation that was learned during the first encounter with the item, when they see the item a second time.

Establishing that activity in particular brain region includes a representation of a temporal code is challenging, as there are many neural signals that show temporal autocorrelation that is not cognitively meaningful. By connecting pattern similarity to performance on a temporal judgment, the authors make meaningful progress with respect to this challenge.

The authors include a variety of tests to refine the specificity of their conclusions. They use statistical models to determine whether different alternative factors could explain the central result (increased pattern similarity across exposures is predictive of later temporal memory). In aggregate, these tests suggest that these hippocampal regions support a representational code that is event-specific, and that this event-specific code can be reactivated when an item is re-encountered. Each of these follow-up tests contributes a piece to the overall picture. I was impressed by the demonstration that the pattern similarity measure was still reliable in CA1 even when compared with a permutation analysis that picked foils that were successfully recognized and which occurred in the same session as the comparison event. All together these tests convincingly rule out many possible alternative explanations that the effect arises from one or another artifact.

On lines 225-228 the authors describe how neither CA1 or ERC show a significant relationship between pattern similarity and recognition confidence. This is described as 'important evidence that [these results] were not a secondary consequence of stronger overall memory for the images'. I am having trouble following the logic of the argument here, it would be helpful if the authors could clarify this point. At the heart of my confusion: Let's say ERC and CA1 showed a significant relationship between pattern similarity and recognition confidence, as well as a significant relationship between pattern similarity and temporal memory precision. Would this undercut the main point of the paper? It seems to me that one could imagine a theory/scenario whereby better temporal memory could support one's recognition judgments. If you remember when something happens you might use that information to inform a recognition decision. So it wasn't clear to me whether I'm missing something, logic-wise. That said, the authors do a good job demonstrating that the temporal effect is reliable even when a statistical model includes recognition performance, so this point is more about the clarity of the text rather than a concern about the results themselves.

Minor comments

There was a paper by Ritchey, Wing, LaBar, and Cabeza (2012) *Cerebral Cortex*, that demonstrated that pattern similarity in hippocampus between encoding and retrieval events was predictive of memory performance. The E2 event is being described as a chance to retrieve the temporal context associated with E1, making the E2 event a retrieval event. It would be useful for the reader if this paper was cited somewhere in the current ms, to make this connection, as the current work builds upon that past work in a meaningful way.

Reviewer #2 (Remarks to the Author):

This manuscript presents unique work that has the potential to substantially advance scientific knowledge. The authors used fMRI scans across unusually long time spans to assess restudy of real-world scenes. They applied cutting edge techniques to determine whether similarity across the information coded in CA1 and entorhinal cortex predicts memory for temporal information. The data support a context reinstatement account and refute an interference account of the nature of memory representations across repetitions of similar events. This is important because we currently have very little neural evidence to indicate if/how prior events are accessed during encoding of similar new events.

There are a few issues I would like to see the authors discuss in more detail:

1. The authors mention findings of time cells in animal studies and acknowledge that the current study does not measure individual cells. The reader might benefit from some additional discussion of the spatial scale at which these analyses are conducted and how it compares to the neuron level.
2. Lines 578-583 indicate that the data were resampled to have both higher spatial and temporal resolution than the original collection parameters. Although a reference is provided to this technique, it was not clear from that reference whether such resampling techniques are known to be robust and resistant to distortion. More information is need on this point. In my opinion, the best possible response would be reporting whether the findings of this study are consistent when conducted on the original data, but I recognize that this is a major undertaking.
3. Related to the above point, some information about additional analyses undertaken that are not reported and/or registration of the intended analysis plan in advance would increase confidence in the results. There are a large number of decision points at which it is not clear that one option is preferable to the other (e.g. "beta version 2 from NSD"; using an HRF that potentially varied for each voxel; dividing continuous temporal judgments via median split). It is critical to explicitly state whether these choices were made in advance of analysis or whether alternative approaches were attempted and produced differing results.

Minor question:

4. I appreciated the detailed descriptions of methods but was confused about the information in lines 547-548 as compared to 563-565. What are the "other images" being described in line 563? Which set was limited to those that received correct responses (143-170 images for each participant)? It seemed that this criterion was already applied in earlier descriptions of image selection.

Reviewer #3 (Remarks to the Author):

In their manuscript, Zou and colleagues present the behavioral and neural results from a large study on temporal memory (i.e., memory for when an event occurred, not just

what occurred.) Their basic results demonstrate that greater neural pattern similarity in CA1 and entorhinal cortex between item presentations, often separated by days to weeks, predicts better temporal memory, but not better recognition memory, which itself was predicted by pattern similarity in parahippocampal cortex.

First of all, this is a fantastic dataset and an important study. Few studies have focused on temporal memory compared to recognition and recall, and we therefore understand much less about the underlying behavioral and neural processes that support successful temporal memory encoding and retrieval. Their data are extremely well suited for exploring temporal memory at more realistic timescales, having collected well-controlled data over many months in each participant. Both the methods and results are clearly written, and the figures are well-designed and easy to interpret, helping the reader interpret the results.

My primary concern is with respect to the narrative and interpretation of the results. The authors propose that temporal memory is supported by temporal context reinstatement at study, but exactly how context reinstatement would support temporal memory requires additional explanation and detail. I follow the discussion in the introduction about repeated item presentations giving rise to either creating a new context or reinstating a prior context, but how either supports good or bad temporal memory is less clear. I think the impact of the paper would be greater if the authors could follow this thought process through all the way. What is the proposed mechanism of temporal memory targeting, and how might this be supported by having a single strengthened context representation, as opposed to distinct representations? And what do you hypothesize is making up the temporal context representation in the first place? Is it other, recently-presented items, as in the temporal context model? If so, perhaps having a single context that is related to the first item presentation supports subsequent temporal memory for when that first item is presented. The proposed mechanism of retrieval would, building off of existing memory models such as BCDMEM and TCM/CMR, involve comparing the overlap of the current test context to the dominant context retrieved when the item was presented at the temporal memory test. The lower the overlap, the more distant the initial item presentation. If instead the item was bound to multiple distinct contexts, there would be less strength of a signal associated with a specific temporal context, making the temporal memory judgment less accurate. Obviously there could be many other possible mechanisms, but stating at least one clearly could help support why the authors see the results they present in the manuscript.

Below I list some comments/suggestions that will hopefully help as the authors revise their manuscript:

- I.48: Although they may code for time, is it not the case that the cells in the ERC behave differently (e.g., with ramping activation patterns as in citation 14), whereas the time cells in the hippocampus fire more in sequences following an event (as in citation 12)? I wouldn't say a major change is needed here, but perhaps they should not all be grouped together as "time cells" for the sake of keeping the clarity in the literature.

- On a related note, it may be worth mentioning that, to date, the timescales over which these cells tracking what and when in the hippocampus and ERC have been observed are not long enough to provide temporal information spanning days, making the present study even more meaningful, but also calling into question to what extent the time cells are relevant to the current study.

- I.132: What was the actual distribution of first times an image was presented over the entire study? I would assume this is skewed heavily towards the first half of the sessions. Perhaps this could be included (in the supplements) as a histogram of counts of first presentations as a function of serial position, combined over all participants. It's important because knowing this distribution will shape how to interpret the temporal memory results.

- Is successful temporal memory dependent on successful identification of an item as new the first time it was presented and then identified as old the second time?

- I don't understand the theory for why temporal memory would be supported by temporal context reinstatement across repetitions of an item. Can you make the connection more explicit (this is likely most important in the introduction)? Please see my suggestions at the top of this review.

- f2C: It looks like there is compression in all the participants' temporal judgments relative to the actual temporal locations. Is there something meaningful about this trend that merits mentioning? Also, is this relationship linear or perhaps is it logarithmic?

- l.174: Technically you shouldn't average correlations, but instead apply a Fisher's transformation to the correlations before averaging. This is mentioned in the methods, but it's not clear in the main text if the transformation was performed for this analysis.

- l.176: What do the correlations between temporal memory precision and the neural similarity metric look like? Is the relationship roughly linear? If so, why do you perform the statistic on the median split of low vs. high temporal memory within each subject instead of simply fitting a mixed-effects regression. Furthermore, why bother with a permutation test when the same could be achieved with a Bayesian regression over individual items? Then you could simply assess whether the group-level posterior slope does not overlap with zero. The issue with the current approach is that you are masking variability in the data by performing the median split, which may actually be decreasing statistical power. This proposed analysis is basically the same as what was presented in Figure 3d. Thus, I'm not sure that including Figure 3c (and the associated analysis) adds anything to the present manuscript. Note, switching to Bayesian statistics is not required, just a suggestion...

- l.183: I'm confused as to why the increased pattern similarity suggests a context reinstatement account and not simply an increased efficacy of encoding, especially at this stage of the narrative. The additional analysis in the paragraph starting at line 223 that describes how pattern similarity did not affect recognition accuracy does potentially help support a mechanism other than encoding efficacy, alone, but I still don't see the direct connection with context reinstatement.

- l.242: As with the earlier analysis, I don't see what is gained from this split half analysis of temporal memory, plotted in Figure 4a and 4b. The regression approach is a far more robust statistical test (though it would be ideal to plot actual fits and data to go along with the beta coefficients).

- l.277: Again, I agree that these results suggest a dissociation between the similarity-based effects that predict temporal as opposed to recognition memory, but the connection to temporal context reinstatement is not clear.

We would like to thank the editor and reviewers for their time and for providing us with detailed and constructive feedback. We have endeavored to address the concerns that were raised by the reviewers. We believe the result is a greatly improved manuscript. Our point-by-point responses are detailed below.

REVIEWER COMMENTS

Reviewer #1 (Remarks to the Author):

Summary 1.0

Zou and colleagues use high-resolution fMRI to examine the reactivation of brain activity patterns related to temporal context memory. This study is unique and innovative in its approach, as it leverages a substantial data set where the same participants were scanned in 30-40 MRI scan sessions each, performing a continuous recognition task. In each session, participants view a series of images, indicating for each image whether they have seen it previously in the experiment. They are tested at the end of the set of sessions using a combination of recognition, frequency, and temporal judgments. This allows the authors to examine temporal order memory on a time-scale previously unheard of in human neuroimaging. The work identifies regions in medial temporal lobe as containing information relevant to the precision of one's memory for past time.

Several key phenomena are demonstrated by the research team, which make this an important contribution to our understanding of the brain circuits important for humans' ability to remember the temporal structure of their past experience. They demonstrate that image-specific brain activity is present in CA1 and ERC, and that increased pattern similarity is associated with better memory for when the item was first encountered in the 40-session sequence (during the final test). This is interpreted as evidence that the participant is reactivating a temporal context representation that was learned during the first encounter with the item, when they see the item a second time.

Establishing that activity in particular brain region includes a representation of a temporal code is challenging, as there are many neural signals that show temporal autocorrelation that is not cognitively meaningful. By connecting pattern similarity to performance on a temporal judgment, the authors make meaningful progress with respect to this challenge.

The authors include a variety of tests to refine the specificity of their conclusions. They use statistical models to determine whether different alternative factors could explain the central result (increased pattern similarity across exposures is predictive of later temporal memory). In aggregate, these tests suggest that these hippocampal regions support a representational code that is event-specific, and that this event-specific code can be reactivated when an item is re-encountered. Each of these follow-up tests contributes a piece to the overall picture. I was impressed by the demonstration that the pattern similarity measure was still reliable in CA1 even when compared with a permutation analysis that picked foils that were successfully recognized and which occurred in the same session as the comparison event. All

together these tests convincingly rule out many possible alternative explanations that the effect arises from one or another artifact.

Response 1.0

We thank the reviewer for their positive assessment of our study and for their insightful comments and suggestions.

Comment 1.1

On lines 225-228 the authors describe how neither CA1 or ERC show a significant relationship between pattern similarity and recognition confidence. This is described as 'important evidence that [these results] were not a secondary consequence of stronger overall memory for the images'. I am having trouble following the logic of the argument here, it would be helpful if the authors could clarify this point. At the heart of my confusion: Let's say ERC and CA1 showed a significant relationship between pattern similarity and recognition confidence, as well as a significant relationship between pattern similarity and temporal memory precision. Would this undercut the main point of the paper? It seems to me that one could imagine a theory/scenario whereby better temporal memory could support one's recognition judgments. If you remember when something happens you might use that information to inform a recognition decision. So it wasn't clear to me whether I'm missing something, logic-wise. That said, the authors do a good job demonstrating that the temporal effect is reliable even when a statistical model includes recognition performance, so this point is more about the clarity of the text rather than a concern about the results themselves.

Response 1.1

We thank the reviewer for raising this important point, which we agree should be clarified. We completely agree that it is possible that better temporal memory could support recognition memory. The point we intended to communicate is that the dissociation of temporal memory from recognition memory is important because it helps constrain mechanistic accounts, not because these forms of memory should or need be dissociable. We recognize that our wording was not clear on this point. In particular, our concern was that temporal information could potentially be inferred from the strength of the memories themselves¹, with stronger memories perceived as more recent^{2,3}. But we agree with the reviewer that the reverse could also be true. The key point is that if CA1 and ERC each showed significant relationships with BOTH temporal memory and recognition confidence, the results (at least on their own) would be compatible with several possible accounts: (1) that these regions fundamentally support temporal memory, which in turn facilitates recognition memory, (2) that these regions fundamentally support recognition memory which in turn is used to infer temporal recency, or (3) that these regions fundamentally support recognition memory and temporal memory via distinct mechanisms. Thus, our finding that CA1 and ERC contributed to temporal memory but not recognition confidence (and that parahippocampal cortex exhibited the opposite pattern) is important because it helps avoid this ambiguity in interpretation. Namely, the observed dissociation suggests a separate mechanism that supports temporal memory decisions, independent from recognition confidence.

We have revised our wording in the Results section in order to be clearer on this point. In particular, we emphasize that these results constrain interpretations and we have removed the word “important” in order to avoid suggesting that it was important to not find a relationship between temporal memory and recognition confidence.

Line 237-242. “These results help constrain accounts of why pattern similarity in CA1/ERC predicted temporal memory precision. Namely, they argue against the possibility that the relationships between CA1/ERC pattern similarity and temporal memory precision were a secondary consequence of overall memory strength for the images. Rather, pattern similarity across exposures in CA1 and ERC specifically predicted better memory for when (in time) images were first encountered.”

Minor comments

Comment 1.2

There was a paper by Ritchey, Wing, LaBar, and Cabeza (2012) *Cerebral Cortex*, that demonstrated that pattern similarity in hippocampus between encoding and retrieval events was predictive of memory performance. The E2 event is being described as a chance to retrieve the temporal context associated with E1, making the E2 event a retrieval event. It would be useful for the reader if this paper was cited somewhere in the current ms, to make this connection, as the current work builds upon that past work in a meaningful way.

Response 1.2

We completely agree and appreciate this suggestion. Indeed, we certainly had this paper in mind but neglected to cite it. We now cite this paper in the Discussion in some new text where we explicitly lay out our argument that E1-E2 pattern similarity reflects retrieval of the original event’s context:

Line 395-399. “From this perspective, our finding that greater pattern similarity across exposures preserved memory for an event’s original temporal context can be explained in terms of the original context representation (elicited during E1) being reinstated and strengthened during subsequent exposures (E2, E3)⁴¹ and ultimately guiding temporal memory judgments at the end of the experiment.”

Reviewer #2 (Remarks to the Author):

Summary 2.0

This manuscript presents unique work that has the potential to substantially advance scientific knowledge. The authors used fMRI scans across unusually long time spans to assess restudy of real-world scenes. They applied cutting edge techniques to determine whether similarity across the information coded in CA1 and entorhinal cortex predicts memory for temporal information. The data support a context

reinstatement account and refute an interference account of the nature of memory representations across repetitions of similar events. This is important because we currently have very little neural evidence to indicate if/how prior events are accessed during encoding of similar new events.

Response 2.0

We thank the reviewer for their positive evaluation of our manuscript and the constructive comments for further improvements.

There are a few issues I would like to see the authors discuss in more detail:

Comment 2.1

1. The authors mention findings of time cells in animal studies and acknowledge that the current study does not measure individual cells. The reader might benefit from some additional discussion of the spatial scale at which these analyses are conducted and how it compares to the neuron level.

Response 2.1

We agree and appreciate this suggestion. We have modified text in the Discussion (see bold text, below) to make explicit the difference in spatial scale while still emphasizing that our analysis approach is well suited to measuring temporal context representations:

Line 428-432. “**Here, we were not able to directly measure or identify time cells because, even with the relatively high spatial resolution of the current fMRI data, each voxel likely pooled across tens or hundreds of thousands of cells.** However, the representation-based analyses we employed are potentially well-suited to capturing gradual changes in ensemble-level context representations^{26,35,36,50–53}.”

Comment 2.2

Lines 578-583 indicate that the data were resampled to have both higher spatial and temporal resolution than the original collection parameters. Although a reference is provided to this technique, it was not clear from that reference whether such resampling techniques are known to be robust and resistant to distortion. More information is needed on this point. In my opinion, the best possible response would be reporting whether the findings of this study are consistent when conducted on the original data, but I recognize that this is a major undertaking.

Response 2.2

We agree that the reason for using the upsampled data was not sufficiently clear in our original manuscript. The decision to use the upsampled data was based on analyses and arguments from the initial data paper (Allen et al., 2021⁴, published in *Nature Neuroscience*) and the decision was made before conducting any of the analyses described in the current manuscript. Specifically, results from the data paper suggest a slight advantage of the upsampled data with limited downside. Importantly, the upsampling involves no additional distortion beyond what is typically done for fMRI pre-processing.

Typically, when motion correction and/or spatial undistortion is applied to fMRI data, there is an interpolation procedure that accommodates the shift in position that has occurred. The upsampling approach we used applies this exact same interpolation procedure, but interpolates onto a fine grid of positions. Ultimately, we have opted against re-doing all of the analyses with the non-upsampled data because (a), as the reviewer notes, this would be a major undertaking, (b), we believe the results would not be substantially different, and (c) the upsampled data are, in principle, better suited for the relatively small regions of interest used here (i.e., hippocampal subfields). Indeed, the main take home from the data paper is that there is some potential upside in terms of recovering finer grained spatial structure, but little to no downside.

We have now clarified our decision in detail in the Methods section, as follows:

Line 577-580. “Informed by the original data paper³⁰, the current study used the upsampled 1.0-mm high-resolution preparation of the NSD data in order to optimally partition the functional data into regions of interest defined using high-resolution anatomical images.”

Comment 2.3

Related to the above point, some information about additional analyses undertaken that are not reported and/or registration of the intended analysis plan in advance would increase confidence in the results. There are a large number of decision points at which it is not clear that one option is preferable to the other (e.g. “beta version 2 from NSD”; using an HRF that potentially varied for each voxel; dividing continuous temporal judgments via median split). It is critical to explicitly state whether these choices were made in advance of analysis or whether alternative approaches were attempted and produced differing results.

Response 2.3

We thank the reviewer for pointing out these issues and apologize for the lack of clarity about the basis for these decisions. All decisions concerning preprocessing and data analysis were made a priori and were motivated by the original NSD data paper and/or the specific questions of interest in the current study. Per the specific issues raised by the reviewer:

- 1) In terms of the version of betas we used (“beta version 2 from NSD”), we apologize that this was not clearly explained/motivated. In the original NSD data paper, three different versions of trial-specific betas were generated and compared. We opted to use the version of the betas (version 2) that allowed for voxel-specific HRFs but that did not assume that the BOLD signal would be consistent across repetitions of an image. Our rationale for allowing for voxel-specific HRFs was that the data paper found that this reduced artifacts. Our rationale for not assuming a consistent response across repetitions was that, for the current study, we were specifically interested in similarity/dissimilarity across repetitions and an assumption of consistent responses was at odds with our research question. Again, these decisions were made before we conducted any of the analyses reported in the current manuscript.

We have revised the Methods section of the manuscript, copied below, so that we no longer describe each of the three models from the NSD data paper (which we feel was unnecessarily confusing). Instead, we note that we used voxel-specific HRFs and now provide a clearer rationale for why did so, emphasizing that this decision was made a priori. We provide a brief overview of how the voxel-specific HRFs were generated, but we refer the reader to the NSD data paper for full details (and make clear that the relevant model from the NSD data paper is “beta version 2”).

Line 581-591. “Parameter estimates (beta weights) reflecting fMRI response amplitudes evoked by each trial were estimated using a general linear model (GLM) approach as described in the NSD data paper³⁰. Notably, our approach (which corresponds to “beta version 2” in the NSD data paper) involved generating voxel-specific hemodynamic response functions (HRFs). Briefly, the pre-processed time-series data were fitted multiple times with a single-trial GLM, each time using a different HRF from a library of HRFs. For each voxel, the HRF that provided the best fit to the data was identified and single-trial betas were then generated using that HRF. Betas were then converted to units of percent BOLD signal change by dividing amplitudes by the mean signal intensity observed at each voxel and multiplying by 100. Our decision to use voxel-specific HRFs was made a priori and was motivated by evidence from the original data paper that it leads to fewer artifacts.”

- 2) In terms of the median split: Consistent with past work using timeline-like tasks⁵, we found that a disproportionate number of responses landed near the middle of the timeline. In other words, responses were compressed. In order to correct for this compression, actual temporal positions and estimated temporal positions (i.e., subjects’ temporal memories) were converted to ranks. This means that the comparison of actual vs. estimated temporal positions was a difference of rank distributions, which is non-normally distributed and not amenable to typical linear regression approaches. These distributions are shown in the figure below, which we now include as a Supplementary Figure:

Supplementary Figure 5. Distribution of individual participant's temporal memory error. Item-wise temporal memory error was calculated using the absolute difference between the ranked estimated temporal position and the ranked actual position. Red line denotes median value of the absolute temporal error. As the resulting distribution was highly non-normal, a median split was performed for all subsequent analyses of temporal error as a factor.

Given the non-normal distributions of ranks, we decided to use a median split to generate “high” vs. “low” precision groups of trials. This decision was made purely on the basis of the qualities of the behavioral data and before we conducted any of the corresponding fMRI analyses.

We have revised the Methods section of the manuscript to clarify the rationale for this decision:

Line 617-621. “Due to the non-normal distribution of temporal error (absolute value of the difference of two rank distributions, see Supplementary Fig. 5), we divided temporal memory trials into ‘high-precision’ and ‘low-precision’ groups by performing a median split of temporal error for each participant. This was an a priori decision (i.e., made before conducting any fMRI analyses).”

Minor question:

Comment 2.4

I appreciated the detailed descriptions of methods but was confused about the information in lines 547-548 as compared to 563-565. What are the “other images” being described in line 563? Which set was

limited to those that received correct responses (143-170 images for each participant)? It seemed that this criterion was already applied in earlier descriptions of image selection.

Response 2.4

We apologize for the confusion. A total of 220 old images were included in the final memory test along with 100 novel images that served as foils. Of the 220 old images, 120 were specifically selected using the three criteria described in lines 539-552, which includes the criterion that they received correct behavioral responses across all three exposures (i.e., correct rejection at E1, then hits at E2 and E3). In other words, these 120 images were included in the final memory test based on the accuracy of responses across the continuous recognition task. In addition to these 120 old images, a second set of 100 old images was included in the final memory test based on semantic properties of those images without consideration of behavioral responses during the continuous recognition task. The rationale for including this second set of images in the final memory test was unrelated to the goals of the current manuscript. Rather, this reflects another potential use of the NSD data set. On the one hand, we could have simply excluded these “extra” images from analysis. However, to increase power, we instead opted to include images from this set so long as they received correct behavioral responses across each exposure during the continuous recognition phase, thereby equating these images (in terms of accuracy during the continuous recognition phase) with the first set of 120 old images. Out of the 220 old images in the final memory test, this yielded 143-170 images for each participant.

We have revised our Methods section in order to clarify this point:

Line 533-564. “The final memory phase included a total of 320 images for each participant: 220 old images and 100 novel images. Old images were selected from the continuous recognition phase that participants completed during fMRI scanning with each image having been presented three times during the continuous recognition phase. Novel images were from the COCO dataset but were not presented during the continuous recognition phase.

The 220 old images included in the final memory test were comprised of two sets of images selected according to different criteria. The first set comprised 120 images that were selected based on behavioral accuracy and the distribution of exposures during the continuous recognition phase. Specifically, these images were selected based on the following three criteria: (1) Each of the 120 images was associated with a correct behavioral response at each exposure during the continuous recognition phase—i.e., correct rejection (E1), hit (E2), and hit (E3). (2) To promote overall temporal memory performance, approximately half of the selected images were associated with a first exposure (E1) that occurred during the last eight scan sessions (for one participant, this was adjusted to the last ten scan sessions in order to have enough trials given their performance in the continuous recognition phase); the other half of the selected images were associated with a first exposure that occurred from the rest of the scan sessions (i.e., earlier sessions). (3) For each half of the images, there was an additional constraint on the spacing between exposures, with one-third of the images having all three exposures within one scan

session, one-third with the last two exposures in the same session, and the rest either with the first two exposures in the same session or with all three exposures across different sessions.

The second set of 100 old images included during the final memory test were selected to maximally span semantic space (see the NSD data paper³⁰ for details), without consideration of behavioral accuracy and distribution of exposures. Briefly, this was done by computing shifted inverse frequency sentence embeddings for the sentence captions, and using a greedy approach to determine the subset of 100 images that maximize the average distance between each image's embedding and its closest neighbor. The motivation for sampling semantic space was unrelated to the goals of the current manuscript. However, in order to increase statistical power, we opted to include images from the second set in our analyses but only for images that were associated with correct behavioral responses for each of the three exposures in the continuous recognition phase (as was the case for the first set). This resulted in a total of 143-170 images that were used for analyses for each participant (see Supplementary Fig. 4 for the distribution of images' first exposures across participants)."

Reviewer #3 (Remarks to the Author):

Summary 3.0

In their manuscript, Zou and colleagues present the behavioral and neural results from a large study on temporal memory (i.e., memory for when an event occurred, not just what occurred.) Their basic results demonstrate that greater neural pattern similarity in CA1 and entorhinal cortex between item presentations, often separated by days to weeks, predicts better temporal memory, but not better recognition memory, which itself was predicted by pattern similarity in parahippocampal cortex.

First of all, this is a fantastic dataset and an important study. Few studies have focused on temporal memory compared to recognition and recall, and we therefore understand much less about the underlying behavioral and neural processes that support successful temporal memory encoding and retrieval. Their data are extremely well suited for exploring temporal memory at more realistic timescales, having collected well-controlled data over many months in each participant. Both the methods and results are clearly written, and the figures are well-designed and easy to interpret, helping the reader interpret the results.

My primary concern is with respect to the narrative and interpretation of the results. The authors propose that temporal memory is supported by temporal context reinstatement at study, but exactly how context reinstatement would support temporal memory requires additional explanation and detail. I follow the discussion in the introduction about repeated item presentations giving rise to either creating a new context or reinstating a prior context, but how either supports good or bad temporal memory is less clear. I think the impact of the paper would be greater if the authors could follow this thought process through

all the way. What is the proposed mechanism of temporal memory targeting, and how might this be supported by having a single strengthened context representation, as opposed to distinct representations? And what do you hypothesize is making up the temporal context representation in the first place? Is it other, recently-presented items, as in the temporal context model? If so, perhaps having a single context that is related to the first item presentation supports subsequent temporal memory for when that first item is presented. The proposed mechanism of retrieval would, building off of existing memory models such as BCDMEM and TCM/CMR, involve comparing the overlap of the current test context to the dominant context retrieved when the item was presented at the temporal memory test. The lower the overlap, the more distant the initial item presentation. If instead the item was bound to multiple distinct contexts, there would be less strong of a signal associated with a specific temporal context, making the temporal memory judgment less accurate. Obviously there could be many other possible mechanisms, but stating at least one clearly could help support why the authors see the results they present in the manuscript.

Response 3.0

We are very happy to hear that the reviewer found our experiment to be “a fantastic dataset and an important study”. We also thank the reviewer for the constructive comments and suggestions. In particular, the comments related to the clarity of our mechanistic interpretations were extremely helpful. Below we outline point-by-point responses to the reviewer’s comments and have revised our manuscript accordingly.

Below I list some comments/suggestions that will hopefully help as the authors revise their manuscript:

Comment 3.1

1.48: Although they may code for time, is it not the case that the cells in the ERC behave differently (e.g., with ramping activation patterns as in citation 14), whereas the time cells in the hippocampus fire more in sequences following an event (as in citation 12)? I wouldn't say a major change is needed here, but perhaps they should not all be grouped together as "time cells" for the sake of keeping the clarity in the literature.

Response 3.1

We thank the reviewer for highlighting this point. We agree it is useful to note this important difference between “time cells” in the hippocampus and “ramping cells” in the ERC. We have now revised our Introduction section accordingly:

Line 48-50. “For example, so-called “time cells” in CA1 and “ramping cells” in ERC have been shown to code for elapsed time in rodents^{12–15}, with similar effects recently observed in the human hippocampus and ERC^{16,17}.”

Comment 3.2

On a related note, it may be worth mentioning that, to date, the timescales over which these cells tracking what and when in the hippocampus and ERC have been observed are not long enough to provide temporal

information spanning days, making the present study even more meaningful, but also calling into question to what extent the time cells are relevant to the current study.

Response 3.2

We appreciate the reviewer for highlighting this point. Indeed, the timescales over which these time-sensitive cells operate are typically very short (seconds). Critically, however, it has been argued that ensembles of these cells allow for coding of temporal information over much longer timescales (hours or days)⁶. The fMRI pattern analyses we used obviously cannot capture individual time cells, but they are closer to the ensemble representations that have been shown in rodents to code temporal information over longer timescales. We have made the following revisions to clarify these important points.

In the Introduction:

Line 50-54. “Importantly, although individual time cells typically operate over very short timescales (seconds), ensembles of time cells may provide temporal context representations that span much longer timescales¹⁸ and allow individual memories to be ‘placed’ in time¹⁹. These temporal context representations are thought to integrate information about temporally adjacent events as well as ongoing internal operations or processes²⁰.”

In the Discussion (also see Response 2.1 for related details):

Line 420-432. “The fact that we specifically identified CA1 and ERC as being important for temporal memory at long timescales—and the implication that these regions supported temporal context reinstatement—is striking in light of accumulating evidence documenting time sensitive cells within rodent CA1 and ERC¹²⁻¹⁴. It has been speculated that ensembles of these cells allow for the coding of gradually-drifting temporal context representations which become bound to individual events¹⁹ and reinstated when events are remembered^{20,27}. Importantly, although individual cells may only operate across very short timescales (seconds), ensemble representations can track temporal information over much longer timescales—from minutes to days^{18,49}. Here, we were not able to directly measure or identify time cells because, even with the relatively high spatial resolution of the current fMRI data, each voxel likely pooled across tens or hundreds of thousands of cells. However, the representation-based analyses we employed are potentially well-suited to capturing gradual changes in ensemble-level context representations^{26,35,36,50-53}.”

Comment 3.3

I.132: What was the actual distribution of first times an image was presented over the entire study? I would assume this is skewed heavily towards the first half of the sessions. Perhaps this could be included (in the supplements) as a histogram of counts of first presentations as a function of serial position, combined over all participants. It's important because knowing this distribution will shape how to interpret the temporal memory results.

Response 3.3

We agree that including the distribution of the first time an image was presented (E1) would be informative. Although first exposures of images by definition occurred earlier in the experiment than later exposures, our method of sampling items to include in the final memory test (see “Stimuli” in the Methods section, lines 543-552) combined with a deliberate effort to spread first exposures out over the course of the experiment (see NSD data paper) minimized a bias toward the first half of the experiment. In fact, the distribution of first exposures for images that were analyzed in the final memory test was not significantly skewed (skewness test: $z = -0.966$, $p = 0.334$). Nevertheless, we agree that this is an important piece of information to include and we have now added the following figure to our Supplementary Figures:

Supplementary Figure 4. Distribution of images' first exposures. Counts of first exposures (E1) for old images included in analyses as a function of session number. Note: counts are summed across participants.

Comment 3.4

Is successful temporal memory dependent on successful identification of an item as new the first time it was presented and then identified as old the second time?

Response 3.4

The reviewer poses an interesting question. In the current study, in order to control for any effects of recognition memory accuracy on temporal memory, we only analyzed items that received correct responses at all three exposures during the continuous recognition phase (see Methods, lines 533-564, for details). However, to address the reviewer's question, we performed an analysis comparing the temporal memory (difference in actual/observed ranked positions) for images that were associated with correct behavioral responses at the first two exposures (i.e., correct rejection at E1 and hit at E2) versus images

for which there was at least one error in the first two exposures (false alarm at E1 and/or miss at E2). Indeed, images that were associated with correct responses at both E1 and E2 were associated with more accurate temporal memory judgments than those that were incorrect at E1 and/or E2 ($t_7 = -2.87$, $p = 0.024$, two-tailed paired-sample t-test). Although we have not added this result to the manuscript (out of concern for overwhelming the reader), we are open to adding this as a Supplemental Figure if the reviewer believes it would be beneficial.

Comment 3.5

I don't understand the theory for why temporal memory would be supported by temporal context reinstatement across repetitions of an item. Can you make the connection more explicit (this is likely most important in the introduction)? Please see my suggestions at the top of this review.

Response 3.5

We thank the reviewer for this very helpful feedback. The logic of our argument rests on two premises: (1) that reinstating the original (E1) temporal context should strengthen the association between the stimulus and the original temporal context, and (2) that a stronger association between the stimulus and the original temporal context should improve temporal memory judgments when that stimulus is encountered yet again at the final temporal memory test. We have revised both the Introduction and Discussion to make this logic clearer and now also provide an explicit description of what a temporal context representation means. There is, however, an additional question of how, exactly, temporal context information is used to support temporal memory judgments. We believe there are likely multiple ways in which this could be achieved. For example, the reviewer suggests that the reinstated context could be compared against the current context with the degree of similarity used to infer recency. This idea makes intuitive sense and there is, in fact, evidence that changes in activity patterns provide a 'read out' of elapsed time⁷. Alternatively, it has been argued that a reinstated context may be systematically compared against sequentially organized contexts from the past and when a 'match' is detected, the corresponding sequential position is used to infer time⁸. Because our findings do not adjudicate between these different accounts, we have opted to simply state that reinstated temporal context information is thought to support temporal memory judgments—and to reference Howard and Eichenbaum (2013)⁹ because this paper provides a nice articulation of this point—without getting into the details of precisely how this is achieved. Interestingly, neuroimaging studies that address temporal context models have overwhelmingly tended to focus on the organization of recall of stimuli (stimulus-stimulus associations) as opposed to actual judgments about temporal memory (stimulus-time associations). Thus, there are still several important open questions in this field.

The specific changes we have made include:

Explicit definition of temporal context representations (Introduction):

Line 50-54. "Importantly, although individual time cells typically operate over very short timescales (seconds), ensembles of time cells may provide temporal context representations that

span much longer timescales¹⁸ and allow individual memories to be ‘placed’ in time¹⁹. These temporal context representations are thought to integrate information about temporally adjacent events as well as ongoing internal operations or processes²⁰.”

Articulating logic of how temporal context reinstatement preserves/strengthens the association between a stimulus and its original context (Introduction):

Line 66-72. “For example, when a familiar movie is on television, this might trigger recall of the original temporal context in which the movie was encountered. Critically, this reinstatement should strengthen the association between the movie and its original temporal context²⁸. According to leading theoretical accounts, a stronger association between a given memory (e.g., the movie) and a particular temporal context (e.g., the movie’s original temporal context) will directly support the ability to place that memory in time²⁹. Thus, in contrast to a context distinctiveness account, a context reinstatement account makes the prediction that, when stimuli are re-encountered, memory for the original temporal context will be preserved to the extent that activity patterns in CA1 and/or ERC are similar to (or reinstate) the activity patterns expressed when the stimulus was first encountered.”

Again clarifying the logic of how our findings align with a context reinstatement account (Discussion):

Line 391-399. “According to temporal context models^{20,27}, context representations—reflected in distributed patterns of neural activity—gradually change over time and are reinstated when a stimulus is subsequently remembered³⁴⁻⁴⁰. This temporal context reinstatement is thought to directly support the ability to remember—or infer—when in time stimuli were previously encountered²⁹. From this perspective, our finding that greater pattern similarity across exposures preserved memory for an event’s original temporal context can be explained in terms of the original context representation (elicited during E1) being reinstated and strengthened during subsequent exposures (E2, E3)⁴¹ and ultimately guiding temporal memory judgments at the end of the experiment.”

Comment 3.6

f2C: It looks like there is compression in all the participants' temporal judgments relative to the actual temporal locations. Is there something meaningful about this trend that merits mentioning? Also, is this relationship linear or perhaps is it logarithmic?

Response 3.6

We thank the reviewer for these questions. The reviewer is correct that there is compression in the temporal judgments as shown in Figure 2c. However, this is due to the fact that these are relationships between two ranked, uniform distributions. In fact, because we ranked the actual and estimated positions, the slope cannot mathematically be greater than 1 (or less than -1). Thus, the observed compression is

more of a mathematical consequence than a psychologically interesting phenomenon. For this reason, we did not comment on or interpret the actual slopes. Importantly, while using ranked positions makes it more difficult to interpret the actual slope, we do believe that ranked positions are the most appropriate measure to use because they correct for overall response biases (namely, participants made more responses near the middle of the timeline, consistent with observations elsewhere using a similar task⁵).

To address the reviewer's question of whether the relationship between actual and estimated positions is linear vs. logarithmic, we fit both a linear and logarithmic function to the raw estimated temporal positions (i.e., not the ranked data) and found a significant correlation in both models, with nearly identical fits (linear model: $r^2 = 0.300$, $p < 0.001$; logarithmic model: $r^2 = 0.290$, $p < 0.001$). However, we have opted against including this data in the manuscript given our concern about (over)interpreting the slope and shape of the function given the responses biases in the task.

Comment 3.7

1.174: Technically you shouldn't average correlations, but instead apply a Fisher's transformation to the correlations before averaging. This is mentioned in the methods, but it's not clear in the main text if the transformation was performed for this analysis.

Response 3.7

We apologize for the confusion. As mentioned in the Methods section, all correlation values were indeed Fisher-transformed before averaging. We have added text to make this explicit.

Line 178-179. "For each region of interest (ROI), we correlated the activity patterns between each pair of exposures of the same image (i.e., $r(E1, E2)$, $r(E2, E3)$, and $r(E1, E3)$). The resulting correlations were then Fisher-transformed prior to statistical testing."

Comment 3.8

1.176: What do the correlations between temporal memory precision and the neural similarity metric look like? Is the relationship roughly linear? If so, why do you perform the statistic on the median split of low vs. high temporal memory within each subject instead of simply fitting a mixed-effects regression. Furthermore, why bother with a permutation test when the same could be achieved with a Bayesian regression over individual items? Then you could simply assess whether the group-level posterior slope does not overlap with zero. The issue with the current approach is that you are masking variability in the data by performing the median split, which may actually be decreasing statistical power. This proposed analysis is basically the same as what was presented in Figure 3d. Thus, I'm not sure that including Figure 3c (and the associated analysis) adds anything to the present manuscript. Note, switching to Bayesian statistics is not required, just a suggestion...

Response 3.8

We thank the review for these suggestions. Before running any of the current analyses, we decided to use a median split for analyzing the temporal memory data (also see Response 2.3.2). The motivation for this

was that the temporal memory precision metric has an atypical (non-normal) distribution because it reflects the absolute values of differences in ranks (see Figure in Response 2.3.2). As such, interpretation of linear fits to this sort of data would not be straight-forward. We believe that the same concerns would apply to interpreting linear Bayesian regression. That said, we believe that our use of logistic regression analyses with a dichotomous precision variable (as we employed for the analyses in Figure 3d) partly addresses the reviewer's concern in that the neural similarity is treated as a continuous predictor variable, even if the temporal precision variable is dichotomous. Ultimately, as the reviewer notes, the greatest concern is that the median split approach may decrease our statistical power or sensitivity given the dichotomous treatment of temporal memory, but, importantly, this concern effectively 'works against' our ability to obtain significant results. Thus, while we recognize there are limitations, we believe our approach is conservative and appropriate given the nature of the data.

Regarding the reviewer's point about the permutation tests used for statistical analyses of the data presented in Figure 3c, we believe that permutation tests are a more appropriate and rigorous way to test these data because the permutation tests do not make any assumptions about the distribution of data (in particular, about the distribution of temporal memory estimates).

Finally, the reviewer is correct that there is some redundancy between Figure 3c and 3d, but we believe that the two panels are complementary in nature and each panel makes an important contribution. The logistic regression analysis which is illustrated in Figure 3d is, arguably, a more sophisticated analysis in part because it allows for us to control for temporal lag (by including it in the model). However, the primary limitation of this analysis is that it does not lend itself to visualization of the actual pattern similarity values. Namely, Figure 3d plots beta values, not pattern similarity. Thus, we feel that Figure 3c is important because it provides a simple visualization of the average pattern similarity values as a function of temporal memory precision (high vs. low). While it is not necessarily surprising that the two analysis approaches provide converging evidence (i.e., similar statistical outcomes), we do believe that including both panels in the figure provides complementary information above and beyond the fact that there is a relationship between the two variables (pattern similarity and temporal memory).

Comment 3.9

1.183: I'm confused as to why the increased pattern similarity suggests a context reinstatement account and not simply an increased efficacy of encoding, especially at this stage of the narrative. The additional analysis in the paragraph starting at line 223 that describes how pattern similarity did not affect recognition accuracy does potentially help support a mechanism other than encoding efficacy, alone, but I still don't see the direct connection with context reinstatement.

Response 3.9

Our intention in mentioning the context reinstatement account at this stage (which, as the reviewer notes is 'early' in the narrative) was to facilitate the reader's understanding of how the results correspond to the account we outline in the Introduction (which is now more clearly articulated). We do appreciate the reviewer's point that this initial result, on its own, is open to other interpretations (like encoding efficacy).

Indeed, we were careful in our choice of language in saying that the results are “consistent with” a context reinstatement account (as opposed to using stronger language). For now, we have opted to leave the sentence intact in the revised manuscript. However, if the reviewer feels it would be better to delete this sentence and to hold off on commenting on consistency with theoretical accounts until later in the manuscript, we are open to doing so.

As the reviewer notes, the dissociation between temporal memory and recognition confidence is a key piece of evidence adjudicating between a context reinstatement account vs. an encoding efficacy account (which we view as the most obvious alternative). However, we acknowledge that the evidence for context reinstatement is indirect. That is, our paradigm and analyses do not afford a direct, unambiguous neural measure of temporal context representations or reinstatement. Indeed, such a measure is difficult to capture because these representations are thought to be a combination of multiple stimulus representations and other fluctuating internal processes. That said, the idea that stimulus repetitions elicit temporal context reinstatement is a well-established *idea* in the field^{10,11} and we believe that a context reinstatement account makes a clear prediction that greater E1-E2 pattern similarity should be associated with better temporal memory for the E1 context. As detailed in other comments, we have revised the manuscript to clarify the logic behind this argument. We would argue that our results are not only consistent with a context reinstatement account, but they provide unique support for this idea because we directly compared temporal memory with recognition confidence and established that these two forms of memory are related to representations in distinct brain regions. Taken together, we are very mindful of overstating the results, but we do feel that our findings are highly consistent with a context reinstatement account and only partially consistent with an encoding efficacy account.

Comment 3.10

1.242: As with the earlier analysis, I don't see what is gained from this split half analysis of temporal memory, plotted in Figure 4a and 4b. The regression approach is a far more robust statistical test (though it would be ideal to plot actual fits and data to go along with the beta coefficients).

Response 3.10

We appreciate the reviewer's point and believe that the logic we endorsed in Response 3.8 would also apply here. Namely, Figure 4a and 4b show actual pattern similarity values (i.e., the y axes) as a function of temporal memory bin (high vs. low precision) whereas this is not shown in Figure 4c and 4d. We agree that showing the logistic fits is a very reasonable alternative, but we ultimately found these to be visually messy and difficult to assess (see below for an example). Thus, we feel that splitting 4a/4b and 4c/4d provides a cleaner/easier way to assess the actual pattern similarity values as a function of temporal memory (4a/4b) and the strength of the relationship between pattern similarity and temporal memory while controlling for temporal lag information (4c/4d).

Comment 3.11

1.277: Again, I agree that these results suggest a dissociation between the similarity-based effects that predict temporal as opposed to recognition memory, but the connection to temporal context reinstatement is not clear.

Response 3.11

Although several of our responses above help clarify how our findings align with a temporal context reinstatement account (see Responses 3.5 and 3.9), we agree with the reviewer that in summarizing this particular result (which now appears on Line 282), the key point is better expressed without reference to temporal context reinstatement. We have revised this sentence (copied below) to instead emphasize the fact that whereas E1-E2 similarity (in CA1 and ERC) was uniquely predictive of *temporal memory*, recognition confidence was actually better predicted by different exposure pairs (E1-E3, but not E1-E2). Thus, we find that temporal memory and recognition confidence were predicted by different brain regions AND by similarity across different exposure pairs. We believe that this dissociation critically informs the interpretation of our results. Namely, we believe that our results, taken together, are more consistent with a temporal context reinstatement account than an encoding efficacy account because the encoding efficacy account would predict that “better encoding” across exposures should lead to better memory overall—including temporal memory *and* recognition confidence.

Line 282. “Along with the results described above, these findings provide a qualitative dissociation between the predictors of temporal memory versus recognition memory. That is, whereas pattern similarity between the first and second exposure of an image was uniquely important for remembering *when* the image was first encountered, it was relatively less important for recognizing *whether* the image was previously encountered.”

Reference:

1. Friedman, W. J. Memory for the time of past events. *Psychol. Bull.* **113**, 44 (1993).
2. Hinrichs, J. V. A two-process memory-strength theory for judgment of recency. *Psychol. Rev.* **77**, 223–233 (1970).
3. Hintzman, D. L. Memory strength and recency judgments. *Psychon. Bull. Rev.* **12**, 858–864 (2005).
4. Allen, E. J. *et al.* A massive 7T fMRI dataset to bridge cognitive neuroscience and artificial intelligence. *Nat. Neurosci.* **25**, 116–126 (2022).
5. Jenkins, L. J. & Ranganath, C. Prefrontal and Medial Temporal Lobe Activity at Encoding Predicts Temporal Context Memory. *J. Neurosci.* **30**, 15558–15565 (2010).
6. Mau, W. *et al.* The Same Hippocampal CA1 Population Simultaneously Codes Temporal Information over Multiple Timescales. *Curr. Biol.* **28**, 1499–1508.e4 (2018).
7. Mankin, E. A. *et al.* Neuronal code for extended time in the hippocampus. *Proc. Natl. Acad. Sci.* **109**, 19462–19467 (2012).
8. Howard, M. W., Viskontas, I. V., Shankar, K. H. & Fried, I. Ensembles of human MTL neurons “jump back in time” in response to a repeated stimulus. *Hippocampus* **22**, 1833–1847 (2012).
9. Howard, M. W. & Eichenbaum, H. The hippocampus, time, and memory across scales. *J. Exp. Psychol. Gen.* **142**, 1211 (2013).
10. Howard, M. W. & Kahana, M. J. A Distributed Representation of Temporal Context. *J. Math. Psychol.* **46**, 269–299 (2002).
11. Sederberg, P. B., Gershman, S. J., Polyn, S. M. & Norman, K. A. Human memory reconsolidation can be explained using the temporal context model. *Psychon. Bull. Rev.* **18**, 455–468 (2011).

Reviewer #1 (Remarks to the Author):

The revision of Zou and colleagues' manuscript was highly responsive to the reviewers concerns, and the manuscript is strengthened as a result of these changes. My own comments were for the most part about how certain results were interpreted or described, and I am certainly satisfied with the revisions addressing these comments.

The changes to the manuscript include two analyses with figures in the supplement, indicating the distribution of images' first exposures, and showing this is even across the sessions. The other shows that individual participants have similar distributions of temporal memory errors. The clarifications made to the text are all beneficial. These include elaborating the reasoning behind why an increase in pattern similarity should predict more precise temporal memory, and an expanded discussion of computational models of temporal context.

All the laudatory comments from my original review stand. This paper makes an important contribution to the cognitive neuroscience literature relevant to theories of episodic memory and event cognition. This is a unique and innovative study with careful analysis methods, that makes appropriately measured, but nevertheless impactful, points about the neural circuitry important for the neural representation of time and temporal judgments.

As a minor potential suggestion (certainly not a concern) I noticed that in the Video attached as part of the supplementary materials, the 'temporal memory test with continuous timeline' looks different in the video than the schematic depiction in Figure 1. Both are perfectly reasonable. In Fig 1c it looks like an arrow is used to mark a point on a session timeline. In the video the mouse has a sort of umbra coming off of it allowing participants to indicate confidence by getting the mouse closer to the timeline. I thought the version in the video was visually striking and thought the authors might consider updating the panel in Fig 1c to look more like that. But perhaps that would require more text than is necessary to get the key aspects of the design across, so I would leave it to the discretion of the authors.

Reviewer #2 (Remarks to the Author):

I served as Reviewer #2 on the previous submission. The authors' responses to my concerns were thoughtful and thorough, as were their responses to the other reviewers. The responses gave me further confidence in the soundness of the study being reported and I remain convinced that it will have a strong impact on the field. I have no further concerns.

Reviewer #3 (Remarks to the Author):

I commend the authors on performing such a thorough revision and careful response to the reviewers.

REVIEWERS' COMMENTS

Reviewer #1 (Remarks to the Author):

The revision of Zou and colleagues' manuscript was highly responsive to the reviewers concerns, and the manuscript is strengthened as a result of these changes. My own comments were for the most part about how certain results were interpreted or described, and I am certainly satisfied with the revisions addressing these comments.

The changes to the manuscript include two analyses with figures in the supplement, indicating the distribution of images' first exposures, and showing this is even across the sessions. The other shows that individual participants have similar distributions of temporal memory errors. The clarifications made to the text are all beneficial. These include elaborating the reasoning behind why an increase in pattern similarity should predict more precise temporal memory, and an expanded discussion of computational models of temporal context.

All the laudatory comments from my original review stand. This paper makes an important contribution to the cognitive neuroscience literature relevant to theories of episodic memory and event cognition. This is a unique and innovative study with careful analysis methods, that makes appropriately measured, but nevertheless impactful, points about the neural circuitry important for the neural representation of time and temporal judgments.

As a minor potential suggestion (certainly not a concern) I noticed that in the Video attached as part of the supplementary materials, the 'temporal memory test with continuous timeline' looks different in the video than the schematic depiction in Figure 1. Both are perfectly reasonable. In Fig 1c it looks like an arrow is used to mark a point on a session timeline. In the video the mouse has a sort of umbra coming off of it allowing participants to indicate confidence by getting the mouse closer to the timeline. I thought the version in the video was visually striking and thought the authors might consider updating the panel in Fig 1c to look more like that. But perhaps that would require more text than is necessary to get the key aspects of the design across, so I would leave it to the discretion of the authors.

We thank the reviewer for this suggestion. As the reviewer notes, changing the panel in Fig 1c would require more text than is necessary to get the key aspects of the design across, we thus opted to keep the current version. Nevertheless, we have modified text in the figure legend to make explicit that Fig 1c is a conceptual illustration of the temporal memory task and more details can be found in Methods and Supplementary Video 1.

Reviewer #2 (Remarks to the Author):

I served as Reviewer #2 on the previous submission. The authors' responses to my concerns were thoughtful and thorough, as were their responses to the other reviewers. The responses gave me further confidence in the soundness of the study being reported and I remain convinced that it will have a strong impact on the field. I have no further concerns.

Reviewer #3 (Remarks to the Author):

I commend the authors on performing such a thorough revision and careful response to the reviewers.